# Computational Algebra with Attention:
# Transformer Oracles for Border Basis Algorithms

**Hiroshi Kera**[*123]     **Nico Pelleriti**[*3]     **Yuki Ishihara**[4]     **Max Zimmer**[3]     **Sebastian Pokutta**[3]
[1]Chiba University     [2]National Institute of Informatics     [3]Zuse Institute Berlin     [4]Nihon University

## Abstract

Solving systems of polynomial equations, particularly those with finitely many solutions, is a crucial challenge across many scientific fields. Traditional methods like Gröbner and Border bases are fundamental but suffer from high computational costs, which have motivated recent Deep Learning approaches to improve efficiency, albeit at the expense of output correctness. In this work, we introduce the ORACLE BORDER BASIS ALGORITHM, the first Deep Learning approach that accelerates Border basis computation while maintaining output guarantees. To this end, we design and train a Transformer-based *oracle* that identifies and eliminates computationally expensive reduction steps, which we find to dominate the algorithm's runtime. By selectively invoking this oracle during critical phases of computation, we achieve substantial speedup factors of up to 3.5x compared to the base algorithm, without compromising the correctness of results. To generate the training data, we develop a sampling method and provide the first sampling theorem for border bases. We construct a tokenization and embedding scheme tailored to monomial-centered algebraic computations, resulting in a compact and expressive input representation, which reduces the number of tokens to encode an $n$-variate polynomial by a factor of $\mathcal{O}(n)$. Our learning approach is data efficient, stable, and a practical enhancement to traditional computer algebra algorithms and symbolic computation.

## 1  Introduction

Polynomial equation systems serve as a fundamental model in a wide range of fields, including dynamical systems (such as those arising from the Lorenz attractor [48] or from digitalized gene-regulatory networks [43, 44]), cryptography where multivariate systems [27, 38] form the basis of signature schemes [67], in data-manifold representation [55], camera-pose estimation [58], or optimization [3, 42]. Solving these systems, even with a small probability of success, reveals critical properties such as stationary points [14] and optimal control strategies [53], highlighting the fundamental importance of polynomial system solving. Many of these applications involve so-called *zero-dimensional systems* with finitely many solutions, which are the focus of this work.

However, solving polynomial systems is notoriously hard: Even for zero-dimensional systems with finitely many solutions, the worst-case complexity is exponential in the number of variates [15, 50] and becomes practically intractable with as few as five variables. A most fundamental tool for their solution is the computation of *Gröbner bases* [10, 13] or *Border bases* [28, 29, 40], the former being interpretable as a nonlinear analogue of Gaussian elimination and the latter being a generalization of the former in the zero-dimensional case.

This work focuses on border basis computation, which is essentially an iterative process that extends a polynomial basis set: at each step, new candidate polynomials are generated from the current basis and *reduced* to determine whether they extend the basis span. Crucially, reducing candidates that

---

[*]Equal contribution.

Correspondence to `kera@chiba-u.jp` or `pelleriti@zib.de`.

fail to extend the basis set is computationally wasteful and leads to long runtime of the algorithm. While several computer-algebraic techniques, including heuristics, address this [24, 29], the advances in the field of Deep Learning (DL) suggest new directions for symbolic computation [4, 35, 41]: A seminal work [54] introduced a Reinforcement Learning approach to optimize candidate polynomial construction in Gröbner basis computation, though their method was limited to binomial Gröbner bases and inputs. A recent work [35] used a Transformer [62] to directly predict Gröbner bases from input systems, but lacks output guarantees - verifying the predicted basis requires a full Gröbner computation, nullifying any efficiency gains.

In this study, we address this problem from the *Algorithms with Predictions* perspective - enhancing algorithms with predictions for better performance, without sacrificing the correctness of the output. To that end, we propose the ORACLE BORDER BASIS ALGORITHM, the first output-certified algorithm for solving zero-dimensional polynomial systems using DL. Precisely, we develop a supervised training framework of a Transformer-based oracle that eliminates unnecessary reduction candidates, thereby significantly improving the efficiency of border basis computation.

**Contributions.** Our contributions can be summarized as follows:

1. **Algorithm for supervised data generation:** To generate the oracle's training data, we develop a sampling method for diverse polynomial sets, realizing the random generation of diverse generators of zero-dimensional ideals. We present the first sampling theorem for border bases and also generalize the *ideal-invariant* transformation theorem [35].

2. **Efficient monomial-level token embedding:** We propose a tokenization and embedding scheme tailored to monomial-centered algebraic computations, resulting in a compact and expressive input representation. For $n$-variate polynomials, it reduces token count by $\mathcal{O}(n)$ and attention memory by $\mathcal{O}(n^2)$ while improving predictive accuracy.

3. **Efficient oracle-guided algorithm with correctness guarantees:** Finally, we propose a Transformer oracle specialized to the final stage of border basis computation, along with an effective heuristic for determining when to invoke the oracle. The resulting ORACLE BORDER BASIS ALGORITHM eliminates unnecessary reduction candidates and significantly improves the efficiency of border basis computation, validated across diverse parameter settings and prediction tasks.

A key insight is that border basis computation proceeds degree-by-degree, enabling us to collect labeled training examples by recording which reductions extend the basis at each degree. In contrast, Gröbner basis computation lacks this natural decomposition, which is why the previous study [54] had to rely on reinforcement learning - a less data-efficient and less stable approach than our supervised learning framework [19, 56, 59, 68]. Since border bases generalize Gröbner bases in the zero-dimensional case, we can efficiently recover Gröbner bases from border bases when they exist.

**Further related work.** Buchberger's algorithm [10] computes Gröbner bases by iteratively forming and reducing S-polynomials until achieving closure under a monomial order [13]. Faugère's F4/F5 algorithms [17, 18] improve upon this via sparse linear algebra and signature-based redundancy detection. Various heuristics [20] help mitigate combinatorial explosion. Border bases do not require a fixed variable ordering and are a term-order-free [9], numerically stable alternative to Gröbner bases, defined relative to a chosen monomial set $\mathcal{O}$ with leading terms on its *border* [28, 29].

Apart from the previously mentioned works on using machine learning for symbolic computation, [26] enhanced cylindrical algebraic decomposition, while [66] improved the stability of Gröbner basis solvers. Our work further connects to the broader field of learning-guided algorithms, where predictions have proven effective at steering heuristic decisions across combinatorial optimization problems [51]. For example, in the context of mixed integer programming, [5] and [36] demonstrated the effectiveness of learning-based approaches to improve branching decisions.

## 2 Preliminaries

We introduce the necessary notation as well as the core concepts of border bases and their computation. For comprehensive treatments, see [28, 29] on border bases and [12] on ideals and polynomials.

An $n$-variate term or monomial $x^{\boldsymbol{a}} := x_1^{a_1} \cdots x_n^{a_n}$ is defined by an exponent vector $\boldsymbol{a} \in \mathbb{Z}_{\geq 0}^n$. We denote by $\mathcal{T}_n$ the set of all such terms and by $K[X] = K[x_1, \ldots, x_n]$ the polynomial ring over a

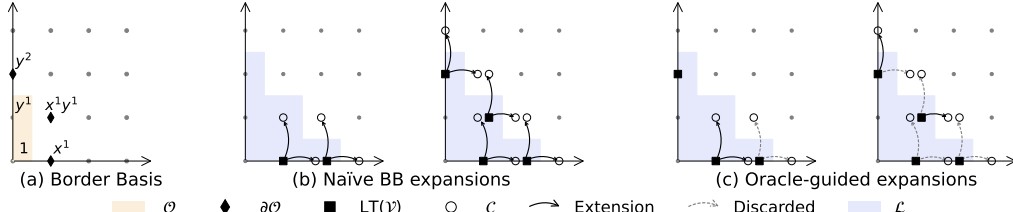

Figure 1: Border basis concepts: (a) A border basis with order ideal $\{1, y\}$ and border terms $\{y^2, xy, x\}$. (b) BBA's iterative expansion of $\mathcal{V}$, showing leading terms: two initial polynomials yield four expansions, then eight more - though only two out of twelve were necessary. (c) The oracle approach achieves the same result with just four targeted expansions.

field $K$. We use $K[X]_{\leq d}$ for its restriction to polynomials of degree at most $d$. We use $\prec$ to denote a term ordering on $K[X]$. The leading term $\mathtt{LT}(f)$ of a polynomial $f$ is its largest term under $\prec$. For a polynomial set $F = \{f_1, \ldots, f_r\}$, $\mathtt{LT}(F) = \{\mathtt{LT}(f_1) \ldots, \mathtt{LT}(f_r)\}$ denotes its leading terms. The set of polynomials $\mathcal{I} = \langle f_1, \ldots, f_r \rangle = \{f_1 h_1 + \cdots + f_r h_r \mid h_1, \ldots, h_r \in K[X]\}$ is called the ideal generated by $F$ in $K[X]$. If $\mathtt{LT}(K[X]) \setminus \mathtt{LT}(I)$ is a finite set, the ideal is called *zero-dimensional*. The cardinality of the set denoted is by $|\cdot|$. For an ideal $\mathcal{I} \subset K[X]$, the quotient ring $K[X]/\mathcal{I}$ denotes the factor ring of $K[X]$ modulo $\mathcal{I}$. Its elements are equivalence classes of polynomials under the relation $f \sim g \iff f - g \in \mathcal{I}$. The symbol $\oplus$ denotes the direct sum of vector spaces: for subspaces $V, W \subset K[X]$, $V \oplus W$ means every element can be uniquely written as $v + w$ with $v \in V$, $w \in W$, and $V \cap W = \{0\}$.

## 2.1 Defining border bases

A border basis of ideal $\mathcal{I}$ is a set of polynomials $\{g_1, \ldots, g_s\}$ that generates $\mathcal{I}$ with specific properties and is defined relative to an *order ideal* $\mathcal{O}$, a subset of monomials that is closed under division.

**Definition 2.1.** A finite set $\mathcal{O} \subset_n$ is an *order ideal* if whenever $t \in \mathcal{T}_n$ divides any $o \in \mathcal{O}$, then $t \in \mathcal{O}$. Its *border* $\partial\mathcal{O} = (\bigcup_{k=1}^{n} x_k \mathcal{O}) \setminus \mathcal{O}$ consists of terms obtained by multiplying elements of $\mathcal{O}$ by variables which are not in $\mathcal{O}$ itself.

**Definition 2.2.** Let $\mathcal{O} \subset \mathcal{T}_n$ be an order ideal with border $\partial\mathcal{O} = \{b_1, \ldots, b_s\}$. A set of polynomials $\mathcal{G} = \{g_1, \ldots, g_s\}$ is an $\mathcal{O}$-*border prebasis* if for all $i = 1, \ldots, s$,

$$g_i = b_i - \sum_{t \in \mathcal{O}} c_t t, \tag{2.1}$$

with $c_t \in K$. If $\mathcal{O}$ forms a basis of $K[X]/\langle G \rangle$, then $G$ is the $\mathcal{O}$-*border basis* of $\langle G \rangle$. In other words, with the span of vector space $\langle \mathcal{O} \rangle_K := \{\sum_{t \in \mathcal{I}} c_t t \mid c_t \in K\}$, we have $K[X] = \mathcal{I} \oplus \langle \mathcal{O} \rangle_K$.

To illustrate, consider a simple polynomial system over $\mathbb{Q}$ with polynomials $x^2 + y^2 - 1$ and $x - 1$. Taking the order ideal $\mathcal{O} = \{1, y\}$, we obtain the border $\partial\mathcal{O} = \{x, yx, y^2\}$. The corresponding border basis is $\{y^2, x - 1, xy - y\}$. It generates the original ideal $\langle x^2 + y^2 - 1, x - 1 \rangle$ and has the required structure: each polynomial's leading term lies in $\partial\mathcal{O}$ while its remaining terms are in $\mathcal{O}$.

## 2.2 Computing border bases

For clarity, we defer the full border basis algorithm to the appendix and focus on its most important component: the *L-stable-span* step [29]. This step is central to the Border Basis Algorithm (BBA), and accelerating it is the primary objective of our work.

The L-stable-span step operates as follows. We begin with a finite set $\mathcal{L} = \{x^\alpha : \|\alpha\|_1 \leq d\}$, which serves as our computational universe of monomials up to degree $d$. We are also given an initial set of polynomials $\mathcal{V}_0 \subseteq \mathrm{span}(\mathcal{L})$, each with a distinct leading term.

At each iteration, we *expand* the current set $\mathcal{V}$ by multiplying every polynomial $v \in \mathcal{V}$ by each variable $x_j$, for $j = 1, \ldots, n$. This produces the set $\mathcal{V}^+ = \{x_j v \mid v \in \mathcal{V}, \ j = 1, \ldots, n\}$. Next,

we perform a linear algebra operation called BasisExtension: we compute a basis for the span of the expanded set, and then restrict this basis to those polynomials whose terms lie within $\mathcal{L}$. If this process does not yield any new elements, the routine terminates.

To clarify the algorithmic process, consider again $\mathcal{V} = \{x-1, x^2+y^2-1\}$ and the initial computational universe $\mathcal{L} = \{1, x, y, x^2, xy, y^2\}$. Multiplying each $v \in \mathcal{V}$ by each variable yields four candidates (cf. Figure 1 b)). After reduction modulo $\mathcal{V}$, two candidates have leading terms in $\mathcal{L}$: $x \cdot (x-1) = x^2 - x$ and $y \cdot (x-1) = yx - y$. Reducing $x^2 - x$ modulo $\mathcal{V}$ gives $x^2 - x - (x^2 + y^2 - 1) = -x - y^2 + 1$. Thus, $\mathcal{V}$ is extended to include $y^2 + x - 1$ and $yx - y$.

Most importantly for our setting, BBA identifies the successful candidates for the border basis only in *hindsight*. To address this, we introduce an oracle that predicts these key polynomials in advance, reducing unnecessary iterations and improving efficiency.

---

**Algorithm 1:** L-Stable Span computation in BBA and OBBA

---

**Input :** Polynomials $\mathcal{V}_0$, universe $\mathcal{L}$;

1   $i \leftarrow 0$
2   **repeat**
3      $\mathcal{C}_i \leftarrow \mathcal{V}_i^+$
4      $\mathcal{C}_i \leftarrow \mathrm{Oracle}(\mathcal{L}, \mathcal{V}_i)$
5      $\mathcal{V}_{i+1} \leftarrow \mathrm{BasisExtension}(\mathcal{V}_i, \mathcal{C}_i, \mathcal{L})$
6      $i \leftarrow i+1$
7   **until** $\mathcal{V}_i = \mathcal{V}_{i+1}$

---

## 3 The Oracle Border Basis Algorithm

In practice, most elements of $\mathcal{C}_i$ are redundant—they vanish after reduction modulo $\mathcal{V}_i$. To eliminate this inefficiency, we introduce an *expansion oracle* that selects a much smaller subset $\mathcal{C}_i \subseteq \mathcal{V}_i^+$; only these polynomials proceed to the reduction step. We refer to the resulting algorithm as the Oracle Border Basis Algorithm (OBBA). The difference to BBA is highlighted in Algorithm 1: instead of setting $\mathcal{C}_i \leftarrow \mathcal{V}_i^+$ as BBA does, we use the oracle $\mathcal{C}_i \leftarrow \mathrm{Oracle}(\mathcal{L}_i, \mathcal{V}_i)$.

The oracle is a lightweight Transformer model (detailed in the next section) that takes a tokenized version of the current computational universe $\mathcal{L}_i$ and generator set $\mathcal{V}_i$ as input, and outputs a set of pairs $(x_\ell, v_m)$, where $x_\ell$ is a variable and $v_m = \mathrm{LT}(p_m)$ is a distinct leading term in $\mathcal{V}_i$. Each pair corresponds to a candidate polynomial $x_\ell p_m \in \mathcal{V}_i^+$.

### 3.1 Termination of the algorithm

We allow the oracle to propose a reduced candidate set up to $k$ times. After the $k$-th invocation, the algorithm defaults to the standard BBA expansion, i.e., $\mathcal{C}_i = \mathcal{V}_i^+$. This strict cap ensures correctness: the oracle can override the vanilla expansion at most $k$ times, after which the algorithm reverts to the standard procedure, preserving both termination and exactness:

**Theorem 3.1.** *OBBA terminates and returns a correct border basis.*

Limiting the oracle to $k$ non-standard expansions prevents repeated expansion in the same direction indefinitely and guarantees termination. For brevity, we present only our main conclusions here and defer detailed empirical and theoretical results to the appendix.

### 3.2 Allocating the $k$ oracle calls

As the algorithm progresses, the generator set $\mathcal{V}$ expands, increasing the number of polynomials to reduce each iteration. Empirically (cf. Table 10), we find that between $70\%$ to $95\%$ of the runtime is spent in the *final stage*, i.e., the iterations that follow the last enlargement of the computational universe $\mathcal{L}$. Since the final iterations dominate the runtime, we aim to spend the oracle budget exclusively there, where every avoided reduction yields the greatest benefit.

A simple yet effective heuristic for recognizing this final stage is to monitor the gap $g := |\mathcal{L}| - |\mathcal{V}|$. When $g$ approaches the size of the (unknown) order ideal $\mathcal{O}$ of the target border basis, no further universe expansion is required (cf. Lemma A.2) and the relative version of this gap, $\frac{|\mathcal{V}|}{|\mathcal{L}|}$ has a strong relationship with the remaining required expansions (cf. Figures 11–13)).

Invoking the oracle too early can be counterproductive: an inaccurate prediction may enlarge $\mathcal{V}$ while still leaving many expansions to run, so the algorithm ends up performing more reductions on a bigger set. Reserving the $k$-oracle calls for the final stage prevents these costs (cf. Theorem A.5).

# 4  Designing the transformer oracle

This section presents the architecture and training of the Transformer oracle

$$\text{ORACLE} : (\mathcal{L}, \mathcal{V}) \mapsto \mathcal{S}, \quad \text{where} \quad \mathcal{L} \subset \mathcal{T}_n, \ \mathcal{V} \subset K[X], \ \mathcal{S} \subset \{x_1, \ldots, x_n\} \times \mathcal{V}. \tag{4.1}$$

We assume a standard encoder–decoder Transformer [62], a general model for sequence-to-sequence tasks. We collect a large number of training samples $((\mathcal{L}, \mathcal{V}), \mathcal{S})$ by running the border basis algorithm. Collecting diverse samples is non-trivial, and we develop new techniques to address this task. Since input sequences to the Transformer oracle are often extremely long (typically tens of thousands of tokens), we also develop methods to substantially reduce their length. All proofs are provided in Appendices B and C.

## 4.1  Dataset generation

Our overall goal is to collect many input–output instances to train the Transformer oracle on, which we could achieve by running the BBA over many sets of polynomials. A crucial challenge is to generate *diverse* sets of polynomials as the following observation highlights. Recall that we are interested in zero-dimensional ideals and let $F \subset K[X]$ be a collection of randomly sampled polynomials (see Appendix E.1 regarding the sampling from $K[X]$). If $|F| < n$, $\langle F \rangle$ is not zero-dimensional. If however $|F| > n$, $\langle F \rangle$ is generally the unit ideal, i.e., $\langle F \rangle = K[X]$. Thus, random sampling only works for $|F| = n$, which is a strong restriction.

We will now address this issue in two steps. First, instead of sampling polynomial systems at random, we propose *border basis sampling* to generate diverse border bases $\{G_i\}_i$—a problem that, to the best of our knowledge, has not been explored in the literature. Secondly, generalizing the results in [35], our *ideal-invariant generator transform* converts each $G_i$ into a non-basis $F_i$ such that $\langle G_i \rangle = \langle F_i \rangle$. This backward approach not only yields a diverse set of polynomial systems with $|F| > n$, but also provides direct control over the complexity of the corresponding border bases, as the sizes and degrees of the border bases (particularly, the order ideals) can be predetermined.

### 4.1.1  Border basis sampling

Our algorithm first samples order ideals, and then constructs border bases supported by them. Recall that a finite set $\mathcal{O} \subset \mathcal{T}_n$ is called an order ideal if for any $t \in \mathcal{O}$, its divisors are all included in $\mathcal{O}$. Thus, for $t \in \mathcal{T}_n$, $\mathcal{O}_t := \{\text{all the divisors of } t\}$ is an order ideal, and the union $\bigcup_{i=1}^q \mathcal{O}_{t_i}$ for $t_1 \ldots, t_q \in \mathcal{T}_n$ is also an order ideal. The latter observation provides a strategy for the sampling of order ideals, requiring refinement of this approach, with the formal algorithm deferred to Appendix B.1.

Now that we can sample order ideals, let $\mathcal{O} = \{o_1, \ldots, o_\nu\}$ be an order ideal and $\partial \mathcal{O} = \{b_1, \ldots, b_\mu\}$ its border. A border *prebasis* $G = \{g_1, \ldots, g_\mu\}$ takes the form $g_i = b_i - \sum_{j=1}^\nu c_{ij} o_j$ for $i = 1, \ldots, \mu$, with arbitrary coefficients $c_{ij} \in K$. To obtain a true border basis, these coefficients must satisfy algebraic conditions ensuring that $\mathcal{O}$ spans the $K$-vector space $K[X]/\langle G \rangle$.

In consequence, random coefficients do not produce border bases. A crucial observation is that the coefficients of $G$ can be readily obtained via simple linear algebra for so-called *vanishing ideals*, which we introduce next.

**Definition 4.1.** Let $P = \{\boldsymbol{p}_1, \ldots, \boldsymbol{p}_\nu\} \subset K^n$ be a set of points. The *vanishing ideal* of $P$ is the set of all polynomials that vanish on $P$, namely, $\mathcal{I}(P) = \{g \in K[X] \mid g(\boldsymbol{p}_i) = 0, \ i = 1, \ldots, \nu\}$.

The following theorem formalizes the construction of a border basis from an order ideal.

**Theorem 4.2.** *Let* $\mathcal{O} = \{o_1, \ldots, o_\nu\}$ *and* $\partial \mathcal{O} = \{b_1, \ldots, b_\mu\}$ *be an order ideal and its border, respectively. Let* $P = \{\boldsymbol{p}_1, \ldots, \boldsymbol{p}_\nu\} \subset K^n$ *be a set of* $\nu$ *distinct points. Let*

$$M(P) := [\partial \mathcal{O}(P) \ \mathcal{O}(P)] := \begin{pmatrix} b_1(\boldsymbol{p}_1) & \cdots & b_\mu(\boldsymbol{p}_1) & o_1(\boldsymbol{p}_1) & \cdots & o_\nu(\boldsymbol{p}_1) \\ \vdots & & \vdots & \vdots & & \vdots \\ b_1(\boldsymbol{p}_\nu) & \cdots & b_\mu(\boldsymbol{p}_\nu) & o_1(\boldsymbol{p}_\nu) & \cdots & o_\nu(\boldsymbol{p}_\nu) \end{pmatrix} \in K^{\nu \times (\mu + \nu)}. \tag{4.2}$$

*If $\mathcal{O}(P)$ is full-rank, the nullspace of $M(P)$ is $\mu$-dimensional and spanned by $\{v_i\}_{i=1}^{\mu}$, where $v_i = (0, \ldots, 1, \ldots, 0, c_{i1}, \ldots, c_{i\nu})^{\top}$, with the first $\mu$ entries being zero except for a 1 in the i-th position. The set $\{g_i = b_i - \sum_{j=1}^{\nu} c_{ij} o_j\}_{i=1}^{\mu}$ is the $\mathcal{O}$-border basis of the vanishing ideal $\mathcal{I}(P)$.*

**Remark 4.3.** Several algorithms [23, 30, 46, 64] can compute a border basis of a vanishing ideal $\mathcal{I}(P)$ from a set of points $P$. However, it only leads to a special type of border bases, which are Zariski-closed (i.e., zero-measure set). Our method covers more general ones, refer to Appendix B.3.

**Remark 4.4.** Non-trivial ideals from random generators are generically *radical*, and any radical ideal $\mathcal{I} = \langle f_1, \ldots, f_r \rangle$ is vanishing ideal $\mathcal{I}(P)$ if the solution set $P$ of $f_1(X) = \cdots = f_r(X) = 0$ is a finite subset of $K^n$. A previous work [35] assumes *ideals in shape position*, and vanishing ideals involve them in this setup. Further generalization to non-radical, positive dimensional ideals is still an open problem.

### 4.1.2 Ideal-invariant generator transform

Now, we are able to obtain a border basis $G = \{g_1, \ldots, g_s\}$ of a vanishing ideal $\mathcal{I}(P)$. Next, we design the following matrix $A \in K[X]^{r \times s}$ to transform a border basis $G = \{g_1, \ldots, g_s\}$:

$$F = AG, \quad \text{s.t.} \ \langle F \rangle = \langle G \rangle. \tag{4.3}$$

This effectively allows us to sample generating sets of $\mathcal{I}(P)$. [35] recently identified a sufficient condition: if $A$ has a left inverse then the ideal-invariant condition $\langle F \rangle = \langle G \rangle$ holds. A subset of such matrices is given by a Bruhat decomposition $A = U_1 P [U_2^{\top} \ O]^{\top}$ with unimodular upper-triangular matrices $U_1 \in K[X]^{r \times r}, U_2 \in K[X]^{s \times s}$ and a permutation matrix $P$ of size $r$. However, the key assumption $|F| \geq |G|$ fails in our case, as typically $|G| \gg n$.

We therefore propose a more general construction of $A$ satisfying the ideal invariance condition.

**Theorem 4.5.** *Let $K$ be a field of characteristic 0, $\mathcal{I}$ a zero-dimensional radical ideal of $K[X]$, and $d$ a positive integer. Let $G \in K[X]^s$ be a generating set of $\mathcal{I}$ and $F = AG \in K[X]^r$ a set of polynomials given by a generic matrix $A \in K[X]_{\leq d}^{r \times s}$.*

1. *If $r \leq n$ and $G$ is a Gröbner basis, we have $\langle F \rangle \neq \langle G \rangle$.*

2. *If $r > n$, we have $\langle F \rangle = \langle G \rangle$.*

This suggests that for $r > n$ and a field $K$ of characteristic 0, the probability that a random $A$ satisfies $\langle AG \rangle = \langle G \rangle$ is almost 1.[2] The probability is also very close to 1 when $K$ is a finite field $\mathbb{F}_p$ with a large prime number $p$.

**Corollary 4.6.** *Let $d$ and $d_{\max}$ be positive integers. Let $K = \mathbb{F}_p$ be a finite field of order $p$ for a prime number $p$ and let $G$ be a subset of $\mathbb{F}_p[X]_{\leq d_{\max}}$ such that $\langle G \rangle$ is a 0-dimensional radical ideal of $\mathbb{F}_p[X]$. Assume $r > n$ and let $\mathcal{G} = \{A \in \mathbb{F}_p[X]_{\leq d}^{r \times s} \mid \langle AG \rangle = \langle G \rangle\}$. Then, a generically sampled $A \in \mathbb{F}_p[X]_{\leq d}^{r \times s}$ satisfies $\langle AG \rangle = \langle G \rangle$ with probability*

$$\Pr(p) = \frac{|\mathcal{G}|}{|\mathbb{F}_p[X]_{\leq d}^{r \times s}|} \geq 1 - \frac{d'}{p} \tag{4.4}$$

*for some positive integer $d'$, which is determined by $d$ and $d_{\max}$, independent of any specific $p$.*

Therefore, for sufficiently large $p$, the success rate of the ideal-invariant transform is almost 1 (note that the vanishing ideal $\mathcal{I}(P)$ is a 0-dimensional radical ideal). See Figure 6 for a numerical experiment. In summary, sampling a random $A$ with more than $n$ rows, which is arguably the simplest approach, works.

## 4.2 Efficient input sequence representation

The input $(\mathcal{L}, \mathcal{V})$ and output $\mathcal{S}$ are respectively regarded as sequences of polynomials (in the fully expanded form), and tokenized into sequences of *tokens*. For example, with $L = \{1, x, y\}$ and

---

[2]Note that in this paper, a *generically (or randomly) sampled* polynomial $f \in K[X]$ refers to a linear combination of all monomials whose degrees are up to an (implicit) upper bound $d$. The combination coefficients are sampled uniformly and independently from the field $K$.

$V = [x + 2, y]$, the input sequence in the infix representation is

$$\texttt{(C1, E0, E0, <sep>, C1, E1, E0, <sep>, C1, E0, E1, <supsep>,}$$
$$\texttt{C1, E1, E0, +, C2, E0, E0, <sep>, C1, E0, E1, <eos>),} \quad (4.5)$$

where `Cn` and `En` represent coefficient and exponent of values $n$. The token `<sep>` separates elements in a set, and `<supsep>` separates sets. The token `<eos>` represents the end of the sequence.

In standard Transformers, the computational cost of self-attention grows quadratically with the input size. The sizes of $\mathcal{L}, \mathcal{V}$ are often large, and $\mathcal{V}$ contains polynomials with many terms. We introduce two methods that, when combined, significantly reduce input size as shown in Figure 2.

**Simplification of in- and output.** We replace $\mathcal{L}$ with its minimal identifying subset $\mathcal{L}' \subset \mathcal{L}$ (cf. Appendix D.3). Since the basis extension step in the BBA primarily relies on leading terms, we truncate each polynomial in $\mathcal{V}$ to its $l$ leading terms, which we found to have minimal impact on the predictive performance of the oracle (cf. Table 1). The target sequence is a list of pairs like $\mathcal{S} = \{(x, v)\}_{x \in X, v \in \mathcal{V}}$. Since the polynomials in $\mathcal{V}$ have mutually distinct leading terms, we can replace each $v \in \mathcal{V}$ with $\texttt{LT}(v)$.

**Monomial embedding.** A fully expanded $n$-variate degree-$d$ polynomial (e.g., $xy + y$ instead of $(x + 1)y$) typically contains on the order of $\binom{n+d}{n}$ monomials. Standard representations (e.g., infix) tokenize each monomial using $n + 1$ tokens—one for the coefficient and $n$ for the exponents. Each monomial is followed by a token like + or `<sep>`, so a polynomial set $F = \{f_1, \ldots, f_r\}$ yields a sequence of the order of $(n + 2)s \cdot \binom{n+d}{n}$. We introduce an efficient embedding scheme for polynomials, representing each monomial with a single token. By combining a monomial and its follow-up token into one vector, this approach removes the $(n + 2)$ factor from the input size.

**Definition 4.7.** (Monomial embedding) Let $\Sigma$ be the set of all tokens. Let $(t, \texttt{<*>})$ be a pair consisting of a monomial $t = cx^{\boldsymbol{a}} \in \mathcal{T}_n$ with coefficient $c \in K$, exponent vector $\boldsymbol{a} \in \mathbb{Z}_{\geq 0}^n$, and a follow-up token $\texttt{<*>} \in \Sigma$. Let $\varphi_c$, $\varphi_e$, and $\varphi_f$ denote embeddings of the coefficient, exponent vector, and follow-up token into a $d$-dimensional space, respectively. The monomial embedding $\varphi_m : \mathcal{T}_n \times \Sigma \to \mathbb{R}^d$ is given by

$$\varphi_m(t, \texttt{<*>}) = \varphi_c(c) + \varphi_e(\boldsymbol{a}) + \varphi_f(\texttt{<*>}). \quad (4.6)$$

Symbolic computations are fundamentally monomial-centric: monomials are compared, added, or divided. Without monomial embedding, attention-based models must connect $(n + 1)$ tokens per monomial; this is reduced to a simple one-to-one mapping, substantially improving success rates in cumulative polynomial product tasks (cf. Table 3). See Appendix D.1 for the exact implementation.

# 5 Experimental results

We empirically evaluate our approach;[3] dataset generation, training details, and additional results are in Appendix E.

## 5.1 Fast Gaussian elimination

As an additional contribution, we present a fast Gaussian elimination (FGE) kernel that replaces the standard elimination in BasisExtension (see Algorithm 1). FGE maintains the active reducer set in a balanced search tree, enabling $O(\log(m))$ look-ups and insertions; finding a reducer with a matching leading monomial is thus logarithmic, not linear. Combined with on-the-fly normalization and an index map for immediate reuse of new reducers, our kernel makes the entire BasisExtension step quasi-linear. This yields an $\approx 10\times$ wall-clock speedup in data generation (cf. Table 7–9), enabling to create an order of magnitude more training samples.

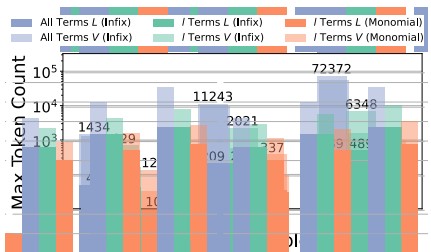

Figure 2: ($\mathbb{F}_{31}$, $k = 5$). The term truncation and monomial embedding significantly reduce input size. See also Figure 8.

---

[3]Our code is available at `https://github.com/HiroshiKERA/OracleBorderBasis`.

Table 1: Evaluation results of Transformer predictions over polynomial ring $\mathbb{F}_{31}[x_1, \ldots, x_n]$. Transformer successfully learns the expansion directions. The input polynomials are truncated to their first $l$ leading terms. The No Expansion Accuracy column shows that the Transformer model can determinate the termination with high accuracy. Refer to Table 5 for the complete version.

| Field | Variables | $l$ | Precision (%) | Recall (%) | F1 Score (%) | No Expansion Acc. (%) |
|---|---|---|---|---|---|---|
| | | 1 | 84.4 | 86.8 | 85.6 | 99.7 |
| | $n = 3$ | 3 | 89.4 | 90.0 | 89.7 | 99.7 |
| | | 5 | 91.6 | 93.2 | 92.4 | 99.7 |
| | | 1 | 90.7 | 91.6 | 91.1 | 98.8 |
| $\mathbb{F}_{31}$ | $n = 4$ | 3 | 92.9 | 93.7 | 93.3 | 98.8 |
| | | 5 | 94.2 | 94.7 | 94.4 | 98.8 |
| | | 1 | 92.7 | 93.1 | 92.9 | 99.6 |
| | $n = 5$ | 3 | 94.3 | 94.6 | 94.4 | 99.6 |
| | | 5 | 94.8 | 95.3 | 95.1 | 99.6 |

Importantly, FGE is orthogonal to the Transformer oracle: while the oracle eliminates unnecessary reductions, FGE accelerates all necessary ones, and their benefits compound.

## 5.2 Learning successful expansions

We begin by demonstrating that the Transformer model can learn to predict successful expansion.

**Dataset.** Datasets were generated as described in Section 4.1, with one million training and one thousand evaluation samples. We set $G \subset K[x_1, \ldots, x_n]_{\leq 2}$ and $A \in K[x_1, \ldots, x_n]^{r \times s}$ for $r \in \{n+1, \ldots, 2n\}$, collecting samples only from the final five expansions of each border basis computation. Each polynomial in $A$ has at most ten terms, sampled as detailed in Appendix E.1. In total, 27 datasets were constructed by varying the number of variables $n \in \{3, 4, 5\}$, coefficient field $\mathbb{F}_p$ with $p \in \{7, 31, 127\}$, and truncation to the $l$ leading terms with $l \in \{1, 3, 5\}$.

**Setup.** Experiments used a standard Transformer with 6 encoder and decoder layers, 8 attention heads, and our monomial embedding. Embedding and feedforward dimensions were $(d_{\mathrm{model}}, d_{\mathrm{ffn}}) = (512, 2048)$. We set dropout to 0.1. The positional embeddings were randomly initialized and trained throughout the epochs. The model was trained for 8 epochs with AdamW [49] ($\beta_1 = 0.9$, $\beta_2 = 0.999$), a linearly decaying learning rate from $10^{-4}$, and batch size 16.

**Results.** Table 1 summarizes our results. The Transformer predicts a set $\mathcal{S} = \{(x_i, v_j)\}$ of expansion direction and target polynomial pairs. For non-empty predictions, we report precision, recall, and F1 score against ground truth; for empty predictions (indicating algorithm termination), we report accuracy. The Transformer consistently learns both expansion directions and target polynomials across all $(n, p, k)$ settings. Notably, it achieves near-perfect accuracy in the No Expansion case, reliably identifying termination and avoiding at least the final unnecessary expansion step. Overall, the performance improves with larger $n$ and $p$, likely due to the higher success rate of the ideal-invariant generator transform (as suggested by Corollary 4.6 and Figure 6). It is also noteworthy that truncation to the $l$ leading terms has only a minor effect on predictive performance, despite significantly reducing input size (cf. Figure 8).

## 5.3 Transformer oracle

We now demonstrate that integrating the Transformer accelerates the improved border basis algorithm. We also evaluate the Transformer oracle's out-of-distribution performance on higher-degree systems, which are more challenging to predict.

### 5.3.1 In-distribution performance

We assess the Transformer oracle's in-distribution performance, i.e., the performance on systems drawn from the same distribution as the training data.

Table 2: Wall-clock runtime in seconds (mean $\pm$ standard deviation over 100 random zero-dimensional systems of total degree $\leq 4$) for five algorithms over polynomial ring $\mathbb{F}_{31}[x_1, \ldots, x_n]$ and variable counts $n = 3, 4, 5$ variables. BBA is the classical border basis algorithm; IBBA is the incremental BBA baseline; OBBA is our oracle-augmented BBA, reducing runtime by about $3\times$ versus IBBA. IBBA+FGE and OBBA+FGE add fast Gaussian elimination (FGE), an orthogonal linear algebra speedup; OBBA+FGE is the fastest, reaching up to two orders of magnitude improvement over IBBA. A $3.5\times$ speedup in terms of unneccessary expansions is shown in the appendix in Table 6.

| Field | $n$ | Baseline | | Ours | | |
| | | BBA | IBBA | OBBA | IBBA+FGE | OBBA+FGE |
| --- | --- | --- | --- | --- | --- | --- |
| | 3 | $0.07 \pm 0.11$ | $0.06 \pm 0.09$ | $0.06 \pm 0.08$ | $0.03 \pm 0.03$ | $\mathbf{0.03} \pm 0.03$ |
| $\mathbb{F}_{31}$ | 4 | $0.46 \pm 0.43$ | $0.35 \pm 0.25$ | $0.22 \pm 0.13$ | $0.09 \pm 0.05$ | $\mathbf{0.09} \pm 0.05$ |
| | 5 | $11.44 \pm 8.25$ | $7.60 \pm 5.13$ | $2.58 \pm 1.37$ | $0.88 \pm 0.49$ | $\mathbf{0.60} \pm 0.32$ |

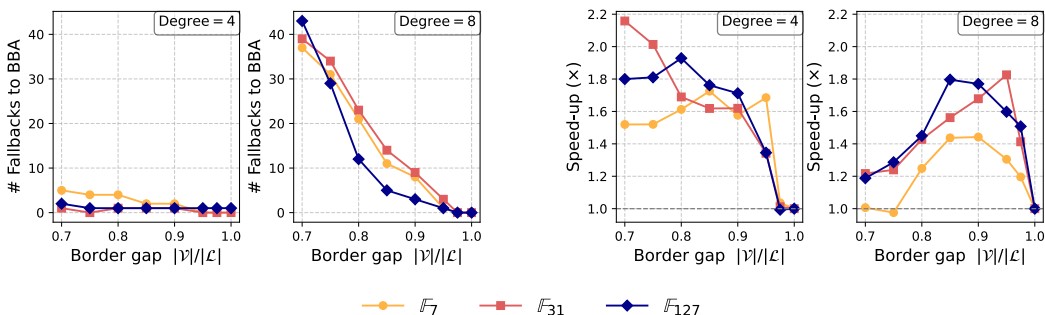

Figure 3: Speed-up of OBBA over IBBA on OOD systems with $n = 4$ variables and increased degree. Each point averages 100 random instances per field. The relative border-gap $\frac{|\mathcal{V}|}{|\mathcal{L}|}$ is the threshold that decides when the oracle is invoked; a ratio of 1 corresponds to IBBA, where the oracle is never used. Although the oracle is trained only on systems of total degree 2 for $n = 4$, it generalizes to degrees 4 and 8, achieving up to $1.8\times$ speed-up even for degree 8. The average runtime for IBBA on the OOD systems is two orders of magnitude higher than for the in distribution case.

**Setup.** We compare five algorithmic variants: the classical BBA, the Improved Border Basis Algorithm (IBBA), IBBA with fast Gaussian elimination (IBBA+FGE), and the oracle-guided versions OBBA and OBBA+FGE. For oracle-augmented variants, the oracle is called at most five times, consistent with our training on the final five expansions. We invoke the oracle at ratios of $0.7, 0.75, 0.8, 0.85, 0.9, 0.95, 0.975$ of the relative border gap $\frac{|\mathcal{V}|}{|\mathcal{L}|}$.

**Results.** Table 2 shows that the oracle-guided algorithm achieves up to a $3\times$ speedup over the state-of-the-art IBBA for systems with five variables, with smaller gains for three and four variables. Notably, none of the oracle-guided variants required reverting to standard iterations, indicating that the Transformer oracle successfully predicts expansion directions and target polynomials for all cases.

### 5.3.2 Out-of-distribution performance

We assess the Transformer oracle's out-of-distribution performance by evaluating on higher-degree systems, which are more challenging for both the oracle to predict and the IBBA to solve. The goal is to test whether the Transformer, trained on easy instances, can generalize to harder ones, where we defer full results to Appendix E.4.

**Setup.** We introduce 27 new datasets (i.e., three variables, three degrees, and three field orders). We increase the total degree of the sampled input system $F$ and transformation matrix $A$ by 1 at each step, limiting ourselves to $\mathbb{F}_7, \mathbb{F}_{31}, \mathbb{F}_{127}$. This produces systems of total degree 3, 4, 6, and 8. To isolate the oracle's impact, we compare IBBA and OBBA without FGE. The average IBBA runtimes (in seconds) on these out-of-distribution instances are: 1.17, 3.21, 20.54, and 40.28 for total degree 3, 4, 6, and 8, respectively, averaged over the three fields.

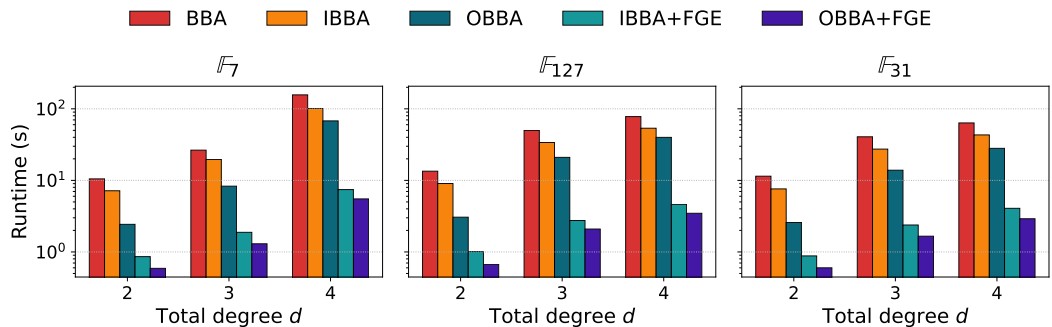

Figure 4: Runtime comparison (log scale) across increasing total degrees $d$ for polynomial systems over $\mathbb{F}_7, \mathbb{F}_{127}$, and $\mathbb{F}_{31}$ with $n = 5$ variables. Bars show mean wall-clock runtimes of 100 instances for two baselines (BBA, IBBA) and three versions of our method (OBBA, IBBA+FGE, and OBBA+FGE). Note the logarithmic y-axis, which reveals that higher-degree out-of-distribution systems are orders of magnitude harder than the degree-2 systems used for training. Despite never being trained on such hard instances, our method,particularly OBBA+FGE, maintains dramatically lower runtimes, showing strong generalization beyond the training distribution to problems that are magnitudes harder.

**Results.** Figure 3 shows that the oracle generalizes to systems with $n = 4$ and total degrees $d = 4$ and $d = 8$, achieving speedup factors of up to $1.8\times$ even for the hardest case ($d = 8$). Despite being trained only on systems of total degree $d = 2$, it generalizes to these substantially more difficult out-of-distribution systems. Figure 4 further illustrates this effect for $n = 5$; OBBA with FGE achieves significant speedups on out-of-distribution systems that are an *order of magnitude harder* than those encountered during training. For the hardest instances, IBBA took on average $101.22$ seconds, while our method reduces this to $5.51$ seconds. Notably, the Transformer was trained only on degree-2 instances, for which IBBA has an average runtime of $7.16$ seconds (cf. Table 7). This highlights the model's strong ability to generalize beyond its training distribution, solving problems that are magnitudes more difficult than those in training. See Appendix E.4 for the full results.

## 6 Conclusion

We introduced a transformer-enhanced border basis algorithm that allows for efficiently solving systems of polynomial equations. Our approach is the first to integrate deep learning into border basis computation, achieving up to 3.5x speedup while fully preserving solution correctness. The development of this oracle-guided method was based on a detailed analysis of algorithmic costs, a new framework for generating diverse training data specific to border bases, and an efficient task-specific polynomial representation, investigated both empirically and theoretically. We believe our work thus provides a practical, data-efficient, and stable enhancement to the symbolic computation toolkit, showcasing a promising way to combine machine learning with established mathematical algorithms.

**Limitations and future work.** This study focuses on 0-dimensional ideals over finite fields. The extension to positive-dimensional and infinite fields is left to our future work. We note that our setup still covers the essential cases. First, any positive-dimensional ideal over finite fields can be reduced to 0-dimensional ones by including the *field equations* (e.g., $x(x - 1)$ and $y(y - 1)$ when $K[X] = F_2[x, y]$), which restricts the solution spaces to $K$. Nevertheless, addressing the sampling of positive-dimensional ideals (and their Gröbner /border bases) is an independent, interesting open problem from an algebraic perspective. Second, all our technical contributions except for those about border bases are compatible with infinite fields; we selected finite fields as a well-known hard case. For example, learning the parity function (i.e., the sum of binary bits with modular 2) has been theoretically and empirically known to be hard to learn [22, 37, 57]. Besides, the finite field case allows us to avoid introducing extra number embedding techniques [11, 69, 70] to address unbounded coefficients. It is also worth noting that infinite-field cases have often been reduced to finite fields through modular techniques [6] for efficient computation. For the practical utility, a scale-up to larger systems is required, although, to the best of our knowledge, our experiments handled the largest and most general class of systems among the related studies [35, 54], see Appendix E.2.

**Acknowledgments.** This research was partially supported by the DFG Cluster of Excellence MATH+ (EXC-2046/1, project id 390685689) funded by the Deutsche Forschungsgemeinschaft (DFG) as well as by the German Federal Ministry of Education and Research (fund number 01IS23025B). Hiroshi Kera was supported by JST PRESTO Grant Number JPMJPR24K4, JST BOOST Program Grant Number JPMJBY24C6, JSPS KAKENHI Grant Number JP23KK0208, Mitsubishi Electric Information Technology R&D Center, and JSPS Program for Forming Japan's Peak Research Universities (J-PEAKS) Grant Number JPJS00420230002. Yuki Ishihara was supported by JSPS KAKENHI Grant Number JP22K13901 and Institute of Mathematics for Industry, Joint Usage/Research Center in Kyushu University (FY2025 Short-term Joint Research "Speeding up of symbolic computation and its application to solving industrial problems 3" (2025a012)). Yuki Ishihara would like to thank Kazuhiro Yokoyama and Yuta Kambe for their helpful comments about backward transforms.

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

# A    Theory of Oracle Border Basis Algorithm

In this section we provide proofs to lemmas and theorem in Section 3. We begin by providing the full algorithm.

---

**Algorithm 2:** Border Bases Algorithm ( BBA  and  OBBA , simplified)

---

**Input :** Polynomial system $F = \{f_1, \ldots, f_r\} \subset K[X]$
1   $d \leftarrow \max\{\deg(f_i) \mid 1 \le i \le r\}; \mathcal{L}_0 \leftarrow \langle \mathcal{T}_n^{\le d} \rangle_k; \mathcal{V}_0 \leftarrow \text{VectorSpaceBasis}(\langle F \rangle_k);$
2   **while** *true* **do**
3      $\mathcal{C}_i \leftarrow \mathcal{V}_i^+$
4      $\mathcal{C}_i \leftarrow \text{Oracle}(\mathcal{L}_i, \mathcal{V}_i)$
5      $\mathcal{V}_{i+1} \leftarrow \text{BasisExtension}(\mathcal{V}_i, \mathcal{C}_i, \mathcal{L}_i)$
6      **if** $\mathcal{V}_{i+1} \ne \mathcal{V}_i$ **then**
7         $\mathcal{V}_i \leftarrow \mathcal{V}_{i+1}$
8         **continue**
9      **end**
10     **if** *not* $\text{BorderBasisCheck}(\mathcal{L}_i, \mathcal{V}_i)$ **then**
11        $d \leftarrow d + 1$
12        $\mathcal{L}_{i+1} \leftarrow \langle \mathcal{T}_n^{\le d} \rangle_k$                         `// Update universe`
13        **continue**
14     **end**
15     **else**
16        $\mathcal{L}_{i+1} \leftarrow \mathcal{L}_i$
17     **end**
18     **break**
19 **end**
20 **return** $\text{FinalReduction}(\mathcal{V}_i, \mathcal{L}_i)$

---

The oracle is restricted to $k$ consecutive non-full expansions, after which we fall back to one final full expansion to ensure correctness. We continue by stating the correctness theorem of the oracle-guided border basis algorithm.

BorderBasisCheck corresponds to the check if the border of the tentative order ideal $\mathcal{L} \setminus \text{LT}(\mathcal{V})$ is already in $\mathcal{L}$. FinalReduction refers to the algorithm with the same name introduced by [28].

**Theorem A.1.** *The oracle-guided border basis algorithm terminates and the output is a border basis.*

*Proof.* There are two techniques to ensure the correctness of the oracle-guided border basis algorithm, one of which will be stated here. After the oracle has been invoked $k$ times, we make one more full expansion. If BasisExtension yields additional generators, we fall back to the standard border basis algorithm. This ensures that the output is a border basis, as we could have started with the current iterate $\mathcal{V}$ and applied the standard border basis algorithm. For this we know correctness, so we are done. $\qquad\square$

## A.1    Alternative termination criteria

To avoid computing a full expansion all the time, we may rely on the *Buchberger Criterion for Border Basis* from [29]. This criteria is often a lightweight alternative to the full expansion. We can directly apply it to the result of the Algorithm 2. If it yields that we have a border basis, we are done. If it fails, we go back to the last iterate $\mathbb{V}_i$ and obtain termination by the standard BBA. In practice, we may combine this approach with a learned heuristic, that decides based on the size of the tentative order ideal, whether a full expansion is necessary or we can directly proceed to check for the Buchberger Criterion.

**Lemma A.2.** *Assume $\mathcal{L}$ needs no further expansion and let $\mathcal{O}$ be the order ideal of the border basis ultimately produced by BBA in Algorithm 1. If $|\mathcal{L}| - |\mathcal{V}| = |\mathcal{O}|$, then no additional expansions are necessary.*

*Proof.* Assume towards a contradiction, a further expansion of $\mathcal{V}$ is required for termination. Thus, in particular it holds we can add at least one element to $\mathcal{V}$, increasing its cardinality by 1. But this contradicts the assumption that $|\mathcal{L}| - |\mathcal{V}| = |\mathcal{O}|$. $\square$

**Setup.** Throughout this section we analyse the *final stage* of computation, where the computational universe $\mathcal{L}$ is fixed. Let $\mathcal{V}_0$ be the first generator set in this stage and assume the vanilla BBA terminates after $T$ full expansions, yielding $\mathcal{V}_T$.

**Definition A.3** (Border distance). The *border distance* between $\mathcal{L}$ and a generator set $\mathcal{V}$ is

$$d(\mathcal{L}, \mathcal{V}) \;=\; \text{"\# of full BBA expansions still required for } \mathcal{V}\text{"}.$$

Hence $d(\mathcal{L}, \mathcal{V}_0) = T$ and $d(\mathcal{L}, \mathcal{V}_T) = 0$.

**Why conditional error?** A $k$-order oracle may cut the border distance by any amount $s \leq \min\{k, d(\mathcal{L}, \mathcal{V}_t)\}$. If $s < \min\{k, d(\mathcal{L}, \mathcal{V}_t)\}$ the oracle has missed *necessary* expansions; if $s = 0$ it has made no progress at all. To separate progress from waste we introduce a conditional error measure.

**Definition A.4** (Conditional prediction error). Invoke a $k$-order oracle at iteration $t \in \{0, \ldots, T\}$. After its $k$ calls let the border distance have dropped by $s$ and denote by $Q$ the *minimal* number of expansions that suffice for an $s$ order oracle to make progress $s$. The *s-progress prediction error* is

$$e(s) \;=\; \sum_{i=0}^{k-1} \big| \operatorname{Oracle}(\mathcal{L}, \mathcal{V}_{t+i}) \big| \;-\; Q.$$

An oracle is ideal if $s = k$ and $e(s) = 0$.

**Benchmark: the optimal expansion sequence.** Let OPT be the cost (number of polynomial reductions) of an *omniscient* algorithm that, starting from $\mathcal{V}_t$, chooses the minimal set of expansions for $s$ subsequent expansions, decreasing the border distance by $s$.

**Theorem A.5** (Cost gap to optimal). *Consider a final-stage instance with universe $\mathcal{L}$, order ideal $\mathcal{O}$, and invoke a $k$-order oracle at iteration $t$. If it achieves progress $s$ with conditional error $e(s)$, then*

$$\operatorname{cost}(\text{OBBA}) \;-\; \text{OPT} \;\leq\; e(s) \;+\; n \max\{T - t - s, 0\} \big( |\mathcal{L}| - |\mathcal{O}| \big).$$

*Proof.* We analyse the overhead by distinguishing two components: The first component is directly given by the prediction error as it determines how many more expansions were done than necessary. By definition, the prediction error is $e(s) = \sum_{i=0}^{k-1} |\operatorname{Oracle}(\mathcal{L}, \mathcal{V}_{t+i})| - Q$ and $Q$ is the cost of the optimal expansion sequence. This covers cost difference for the first $s$ iterations. Suppose now, that $T - t - s = 0$. Then, the algorithm terminates and no further error is incurred. Otherwise, we might incur an additional cost for having expanded the generator set $\mathcal{V}$ too much without reducing the border distance beyond $s$. The size of constructed generator set is at most $|\mathcal{L}| - |\mathcal{O}|$. We know that from the optimal $s$ expansions, after $k - s$ expansions, the set $\mathcal{V}$ has at least the size of the generator set that was predicted by the oracle. Therefore, there are at most $T - t - s$ remaining iterations that incur the higher cost which are upper bounded by $n(|\mathcal{L}| - |\mathcal{O}|)$ times per expansion. $\square$

The overhead therefore consists of two terms: the first grows linearly with the prediction error, while the second reflects a hidden cost for making only progress $s$. Specifically, if the oracle expands the generator set $\mathcal{V}$ too much without reducing the border distance beyond $s$, we incur additional cost in the following $T - t - s$ iterations. This theorem is practically relevant, as it further justifies the use of the $k$-order oracle in the final stage of computation. Being conservative with the use of the oracle (i.e. such that $T$ is smaller than $k$), avoids the worst case additional cost.

# B Border basis sampling

## B.1 Order ideal sampling

We introduced the overview of order ideal sampling in Section 4.1.1. This section presents the formal algorithm of the order ideal sampling given . The extension to general $n$-dimensional case is based on the same idea, but it requires a more careful formalization than one may expect.

Recall our idea. A finite set of terms $\mathcal{O}$ is called an order ideal if for any $t \in \mathcal{O}$, its divisors are also all included in $\mathcal{O}$. Importantly, for any terms $t_1, t_2 \in \mathcal{T}_n$, the term set $\mathcal{O}_{t_i} := \{\text{all the divisors of } t_i\}$ is an order ideal for each $i$, and their union $\mathcal{O}_{t_1, t_2} = \mathcal{O}_{t_1} \cup \mathcal{O}_{t_2}$ is also an order ideal. We sample order ideals based on this observation. During the iterative process, we maintain a list $Q$ of *cells*. At each iteration, we pop out a cell $C$ from $Q$, and split it to generate new cells. The cells with sufficient size are appended to $Q$, and this process iterates until $Q$ becomes empty.

We now formalize this idea. From now on, we work on the exponent vectors of terms. We denote a vector $\boldsymbol{v} \in \mathbb{N}^n$ with a replacement of the $i$-th entry with a scalar $p \in \mathbb{N}$ by $\boldsymbol{v}^{[i \leftarrow p]}$. A cell is a tuple of $n$ segments and an intersecting point $p \in \mathbb{N}^n$.

**Definition B.1.** Let $\boldsymbol{a}, \boldsymbol{b} \in \mathbb{N}^n$. The following set is called the *segment* of them.

$$\Delta(\boldsymbol{a}, \boldsymbol{b}) = \left\{ \boldsymbol{r} := (r_1, \ldots, r_n)^\top \in \mathbb{N}^n \mid \min(a_i, b_i) \leq r_i \leq \max(a_i, b_i), \ i = 1, \ldots, n \right\}. \quad \text{(B.1)}$$

The maximum point of the segment is defined by $\overline{\Delta(\boldsymbol{a}, \boldsymbol{b})} := (\max(a_1, b_1), \ldots, \max(a_n, b_n))^\top$, and the minimum point $\underline{\Delta(\boldsymbol{a}, \boldsymbol{b})}$ is defined similarly.

**Definition B.2.** Let $\Delta_1, \ldots, \Delta_n$ be some segments such that $\underline{\Delta_1} = \cdots = \underline{\Delta_n} = \boldsymbol{l}$. The *cell* is a tuple $C = (\{\Delta_i\}_{i=1}^n, \boldsymbol{l})$, and $\boldsymbol{l}$ is called the *intersecting point*. Further, the *maximum point* of the cell is $\boldsymbol{u} = (u_1, \ldots, u_n)^\top \in \mathbb{N}^n$, where $u_i = \min(\overline{\Delta_{1i}}, \ldots, \overline{\Delta_{i-1\,i}}, \overline{\Delta_{i+1\,i}}, \ldots \overline{\Delta_{ni}})$ for $i = 1, \ldots, n$. The cell is called *valid* if $\boldsymbol{l}$ and $\boldsymbol{u}$ has at least two different entries, and $\max_{i \in \{1,\ldots,n\}} u_i - l_i \geq 2$.

Let $\boldsymbol{d} = (d_1, \ldots, d_n)^\top \in \mathbb{N}^n$ be the vector of the maximum degree of variables. The initial cell is defined by $C_0 = (\{\Delta(\boldsymbol{0}, d_i \boldsymbol{e}_i)\}_{i=1}^n, \boldsymbol{0})$, and let $Q = [C_0], R = [\,]$ be lists. Then, we repeat the following until $Q$ becomes empty or the number of iterations reaches the predesignated limit (if any).

1. Select a cell $C = (\{\Delta_i\}_{i=1}^n, \boldsymbol{l}) \in Q$ and remove it from the list.

2. Sample a vector $\boldsymbol{p} = (p_1, \ldots, p_n)^\top \in \mathbb{N}^n$ with $p_i \in [l_i, u_i]$ for $i = 1, \ldots, n$.

3. Append a tuple $(\boldsymbol{l}, \boldsymbol{p})$ to $R$.

4. For $i = 1, \ldots, n$, obtain a new cell $C_i^{\text{new}} = \left( \{\Delta_j^{\text{new}}\}_{j=1}^n, \boldsymbol{l}^{[i \leftarrow p_i]} \right)$, where $\Delta_j^{\text{new}} = \Delta\left( \underline{\Delta_j}^{[j \leftarrow p_j]}, \overline{\Delta_j} \right)$ for $j \neq i$ and otherwise $\Delta_j^{\text{new}} = \Delta(\boldsymbol{l}, \boldsymbol{p})$. Append $C_i^{\text{new}}$ to $Q$ if it is a valid cell.

We then have an order ideal

$$\mathcal{O} = \left\{ x^{\boldsymbol{a}} \mid \boldsymbol{a} \in \bigcup_{(\boldsymbol{l}, \boldsymbol{p}) \in R} \bigtimes_{i=1}^{n} [l_i, p_i] \right\}. \quad \text{(B.2)}$$

**Theorem B.3.** *The order ideal sampling algorithm terminates within a finite number of steps.*

*Proof.* **Termination.** We prove that the order ideal sampling algorithm terminates after a finite number of iterations.

Let $C = (\{\Delta_i\}_{i=1}^n, \boldsymbol{l})$ be a cell, and denote its maximum point as $\boldsymbol{u} = (u_1, \ldots, u_n)$. In each iteration, the algorithm pops one such cell from the queue $Q$, samples a point $\boldsymbol{p} \in \bigtimes_{i=1}^{n}[l_i, u_i]$, and generates at most $n$ new valid cells $C_1^{\text{new}}, \ldots, C_n^{\text{new}}$.

**Case 1: $\boldsymbol{p} \neq \boldsymbol{l}$.** For any $i$, the new cell $C_i^{\text{new}}$ contains at least one segment that is strictly smaller than the corresponding one in $C$. In particular, its $i$-th segment is $\Delta_i^{\text{new}} = \Delta(\boldsymbol{l}, \boldsymbol{p})$, which has length zero in coordinate $j$ if $p_j = l_j$, and is strictly shorter otherwise. Therefore, the total number of exponent vectors in $C_i^{\text{new}}$ is strictly less than that in $C$.

**Case 2: $\boldsymbol{p} = \boldsymbol{l}$.** For any $i$, the new cell $C_i^{\text{new}}$ becomes invalid. Indeed, let $\boldsymbol{u}' = (u_1', \ldots, u_n')$ be its maximum point. Since $\Delta_i^{\text{new}} = \boldsymbol{l}$ by construction, and for all $k \neq i$ the $k$-th segment has lower and upper bounds both equal to $l_k$, we have $u_k' = l_k'$ for all $k$. This violates the validity condition, which requires that $\boldsymbol{l} \neq \boldsymbol{u}$ and $\max_i(u_i - l_i) \geq 2$.

Therefore, in either case, the number of valid cells does not increase: Case 1 generates strictly smaller valid cells, and Case 2 generates no valid cells. Moreover, since the exponent space $[0, d_1] \times \cdots \times$

$[0, d_n]$ is finite, and each valid cell corresponds to a distinct subregion defined by its segments, the total number of distinct valid cells is finite.

Hence, the queue $Q$ becomes empty after finitely many iterations, and the algorithm terminates.

$\square$

**Empirical results.** Figure 5 shows a gallery of randomly sampled order ideals in the two-dimensional case. As can be seen, it involves diverse order ideals.

## B.2 Border basis construction from order ideals: Proof of Theorem 4.2

**Definition B.4.** Given a set of points $P = \{\boldsymbol{p}_1, \ldots, \boldsymbol{p}_\nu\} \subset K^n$, with gentle abuse of notation, the *evaluation vector* of a polynomial $h \in K[X]$ is defined by

$$h(X) = (h(\boldsymbol{p}_1) \; \cdots \; h(\boldsymbol{p}_\nu))^\top \in K^\nu.$$

For a set of polynomials $\mathcal{H} = \{h_1, \ldots, h_\mu\} \subset K[X]$, the *evaluation matrix* is defined as

$$\mathcal{H}(P) = (h_1(P) \; \cdots \; h_\mu(P)) \in K^{\nu \times \mu}.$$

The proof of Theorem 4.2 immediately follows from the following lemma.

**Lemma B.5.** *Let $\mathcal{O} = \{o_1, \ldots, o_\nu\} \subset \mathcal{T}_n$ and $G \subset K[X]$ be an order ideal and its $\mathcal{O}$-border prebasis, respectively. Let $P \subset K^n$ be a set of $s$ points. If the following hold,*

1. $\langle o_1(P), \ldots, o_\nu(P) \rangle_k = K^\nu$.

2. $G \subset \mathcal{I}(P)$.

*then, $G$ is the $\mathcal{O}$-border basis of the vanishing ideal $\mathcal{I}(P)$.*

*Proof.* We first show that $G$ is the $\mathcal{O}$-border basis of $\langle G \rangle$. The set $G$ is a border prebasis and (obviously) generates $\langle G \rangle$, so we only need to show that $K[X] = \langle G \rangle \oplus \langle \mathcal{O} \rangle_k$ holds. The border basis division algorithm [28] allows us to represent any polynomial $f \in K[X]$ as $f = g + h$ with some $g \in \langle G \rangle$ and $h \in \langle \mathcal{O} \rangle_k$, which implies $K[X] = \langle G \rangle + \langle \mathcal{O} \rangle$. Note that this algorithm only requires $G$ to be a border prebasis (not necessarily a border basis). From the first assumption, $o_1(P), \ldots, o_\nu(P)$ form a basis of $K^\nu$, and thus, $h \in \langle \mathcal{O} \rangle_K$ has $h(P) = \boldsymbol{0}$ only when $h = 0$. Noting that $\forall g \in \langle G \rangle, g(P) = \boldsymbol{0}$ from the second assumption, we have $\langle G \rangle \cap \langle \mathcal{O} \rangle_K = \{0\}$. Thus, we have $K[X] = \langle G \rangle \oplus \langle \mathcal{O} \rangle$, and $G$ is the $\mathcal{O}$-border basis of $\langle G \rangle$. Furthermore, the fact that $\mathcal{I}(P) \cap \langle \mathcal{O} \rangle_k = \{0\}$ implies $\mathcal{I}(P) \subset \langle G \rangle$. With the second assumption, we have $\mathcal{I}(P) = \langle G \rangle$, and thus, $G$ is the $\mathcal{O}$-border basis of $\mathcal{I}(P)$. $\square$

**Theorem 4.2.** *Let $\mathcal{O} = \{o_1, \ldots, o_\nu\}$ and $\partial \mathcal{O} = \{b_1, \ldots, b_\mu\}$ be an order ideal and its border, respectively. Let $P = \{\boldsymbol{p}_1, \ldots, \boldsymbol{p}_\nu\} \subset K^n$ be a set of $\nu$ distinct points. Let*

$$M(P) := [\partial \mathcal{O}(P) \; \mathcal{O}(P)] := \begin{pmatrix} b_1(\boldsymbol{p}_1) & \cdots & b_\mu(\boldsymbol{p}_1) & o_1(\boldsymbol{p}_1) & \cdots & o_\nu(\boldsymbol{p}_1) \\ \vdots & & \vdots & \vdots & & \vdots \\ b_1(\boldsymbol{p}_\nu) & \cdots & b_\mu(\boldsymbol{p}_\nu) & o_1(\boldsymbol{p}_\nu) & \cdots & o_\nu(\boldsymbol{p}_\nu) \end{pmatrix} \in K^{\nu \times (\mu + \nu)}. \quad (4.2)$$

*If $\mathcal{O}(P)$ is full-rank, the nullspace of $M(P)$ is $\mu$-dimensional and spanned by $\{\boldsymbol{v}_i\}_{i=1}^\mu$, where $\boldsymbol{v}_i = (0, \ldots, 1, \ldots, 0, c_{i1}, \ldots, c_{i\nu})^\top$, with the first $\mu$ entries being zero except for a 1 in the $i$-th position. The set $\{g_i = b_i - \sum_{j=1}^\nu c_{ij} o_j\}_{i=1}^\mu$ is the $\mathcal{O}$-border basis of the vanishing ideal $\mathcal{I}(P)$.*

*Proof of Theorem 4.2.* The existence of such basis set $\{\boldsymbol{v}_i\}_{i=1}^\mu$ readily follows from the assumption that $\mathcal{O}(P)$ is full-rank. The second statement follows from Lemma B.5. $\square$

## B.3 Comparison with Buchberger–Möller-family algorithms

The Buchberger–Möller (BM) algorithm [2, 52] is a method that takes as input a set of points $P$ and computes a Gröbner basis of the vanishing ideal $\mathcal{I}(P)$. This algorithm has been extensively studied and adapted to a variety of scenarios, not only in computer algebra [1, 2, 9, 16, 23, 29, 30, 34, 46], but also in machine learning [25, 31–33, 39, 47, 55, 64, 65].

Border basis variants of the BM algorithm allow us to sample border bases from randomly generated sets of points and a term order $\prec$. However, only a restricted class of border bases can be sampled in this way. Recall that a border basis $G = \{g_1, \ldots, g_s\}$ is associated with an order ideal $\mathcal{O}$, and for each border term $b_i \in \partial\mathcal{O}$, the corresponding border basis polynomial takes the form

$$g_i = b_i - \sum_{t \in \mathcal{O}} c_t t \in K[X], \quad \text{where } c_t \in K. \tag{B.3}$$

**Remark B.6.** The $\mathcal{O}$-border basis $G = \{g_1, \ldots, g_s\}$ produced by BM-type algorithms with a term order $\prec$ is a special case satisfying $\mathrm{LT}(g_i) = b_i$ for all $i = 1, \ldots, s$. That is, the coefficients $c_t$ vanish for any $t \in \mathcal{O}$ such that $t \succ b_i$. Due to this algebraic constraint, the collection of such border bases forms a Zariski-closed (i.e., measure-zero) subset in the set of all border bases.

Such special border bases can also be characterized as those admitting some order ideal $\mathcal{O}$ for which there exists an "$\mathcal{O}$-Gröbner basis" $G'$ generating the same ideal, satisfying $\mathrm{LT}(K[X]) \backslash \mathrm{LT}(\langle G' \rangle) = \mathcal{O}$. While border bases are defined with respect to an order ideal $\mathcal{O}$, Gröbner bases are defined with respect to a term ordering. The former is a strictly more general notion in the zero-dimensional case. For instance, one can find examples of border bases for which no corresponding Gröbner basis exists [9].

# C Backward transform

## C.1 Proof of Theorem. 4.5

**Theorem 4.5.** *Let $K$ be a field of characteristic $0$, $\mathcal{I}$ a zero-dimensional radical ideal of $K[X]$, and $d$ a positive integer. Let $G \in K[X]^s$ be a generating set of $\mathcal{I}$ and $F = AG \in K[X]^r$ a set of polynomials given by a generic matrix $A \in K[X]^{r \times s}_{\leq d}$.*

*1. If $r \leq n$ and $G$ is a Gröbner basis, we have $\langle F \rangle \neq \langle G \rangle$.*

*2. If $r > n$, we have $\langle F \rangle = \langle G \rangle$.*

*Proof.* We write an outline of the proof (see below for details). When $r < n$, the codimension of $\langle AG \rangle$ is less than $n$ and thus $\langle F \rangle \neq \langle G \rangle$ holds for any $A \in K[X]^{r \times s}_{\leq d}$. When $r \geq n$, $\langle F \rangle = \langle G \rangle \cap J$ for some ideal $J$. If $r = n$, then $J \neq K[X]$ for a generic matrix $A \in K[X]^{r \times s}_{\leq d}$ and thus $\langle F \rangle \neq \langle G \rangle$. If $r \geq n$, then $J = K[X]$ for a generic matrix $A \in K[X]^{r \times s}_{\leq d}$ and thus $\langle F \rangle = \langle G \rangle$. $\qquad\square$

To prove $\langle AG \rangle = \langle G \rangle$ for a generic matrix $A \in K[X]^{r \times s}_{\leq d}$, we consider a parametric matrix $\overline{A}$. Let $h_{ij} = \sum_{|\alpha| \leq d} a_{\alpha,ij} X^\alpha$ be a parametric polynomial of total degree $d$ with parameters $a_{\alpha,ij}$ and $\overline{A} = (h_{ij})$ an $r \times s$ matrix with $h_{ij}$ as $(i, j)$ entry. Let $\mathcal{A} = \{a_{\alpha,ij}\}$ be the set of all parameters of $\overline{A}$ and $D$ the cardinality of $\mathcal{A}$. For a generating set $G = \{g_1, \ldots, g_s\}$ of a zero-dimensional radical ideal $\mathcal{I}$, let $f_j = \sum_{k=1}^{s} h_{jk} g_k$ for $j \in \{1, \ldots, r\}$, that is, $\overline{A}G = \{f_1, \ldots, f_r\}$. For a subset $S$ in a ring $R$, we also write $\langle S \rangle_R$ when we emphasize that $\langle S \rangle$ is an ideal of $R$. Obviously, $\langle \overline{A}G \rangle_{K[\mathcal{A}, X]} \subset \langle G \rangle_{K[\mathcal{A}, X]}$. For a point $q \in K^D$, we denote by $\sigma_q$ the substitution map from $K[\mathcal{A}, X] \to K[X]$, where $\sigma_q(f(\mathcal{A}, X)) = f(q, X)$. Then, $A := \sigma_q(\overline{A}) := (\sigma_q(h_{ij})) \in K[X]^{r \times s}_{\leq d}$ is a generic matrix for a generic $q \in K^D$. Thus, it is enough to show that there exists a dense set $C$ of $K^D$ such that $\langle \sigma_q(\overline{A})G \rangle = \langle G \rangle$ for any $q \in C$. As $\langle \sigma_q(\overline{A})G \rangle \subset \langle G \rangle$ always holds, the inverse inclusion implies the equality. First, we recall some fundamental notions of Commutative Algebra as follows.

**Definition C.1** (Primary Decomposition)**.** Let $\mathcal{I}$ be an ideal. A finite set of primary ideals $\{Q_1, \ldots, Q_l\}$ is a primary decomposition of $\mathcal{I}$ if $\mathcal{I} = Q_1 \cap \cdots \cap Q_l$. A primary decomposition $\{Q_1, \ldots, Q_l\}$ of $\mathcal{I}$ is said to be minimal if the length $l$ is minimum among all primary decompositions of $\mathcal{I}$. For a minimal primary decomposition $\{Q_1, \ldots, Q_l\}$ of $\mathcal{I}$, each element $Q_i$ is called a primary component of $\mathcal{I}$. In addition, a primary component $Q_i$ is said to be isolated if $\sqrt{Q_i} \not\subset \sqrt{Q_j}$ for any $j \neq i$, where $\sqrt{Q_i}$ is the radical ideal of $Q_i$.

**Remark C.2** ([7], Corollary 4.11)**.** The isolated primary components of $\mathcal{I}$ are uniquely determined from $\mathcal{I}$. In other words, they are independent of a particular primary decomposition of $\mathcal{I}$.

Independent sets are a useful tool to compute the dimension of a polynomial ideal.

**Definition C.3** (Independent Set)**.** Let $U$ be a subset of $\mathcal{A} \cup X$. For an ideal $\mathcal{I}$ of $K[\mathcal{A}, X]$, $U$ is called an independent set mod $\mathcal{I}$ if $\mathcal{I} \cap K[U] = \{0\}$.

**Remark C.4** ([21], Theorem 3.5.1 (6))**.** The cardinality of an independent set mod $\mathcal{I}$ is less than or equal to the dimension of $\mathcal{I}$. In other words, if $|U| > \dim(\mathcal{I})$ then $U$ is not an independent set mod $\mathcal{I}$.

Let $\iota$ be the inclusion map $\iota : K[\mathcal{A}, X] \to K(X)[\mathcal{A}]$ such that $\iota(f) = f$. For a prime ideal $\mathcal{I}$ of $K(X)[\mathcal{A}]$, the inverse image $\iota^{-1}(\mathcal{I}) = \mathcal{I} \cap K[\mathcal{A}, X]$ is also prime as follows.

**Lemma C.5** ([8], Lemma 1.123)**.** *Let $\mathcal{I}$ be a prime ideal of $K(X)[\mathcal{A}]$. Then, $\mathcal{I} \cap K[\mathcal{A}, X]$ is a prime ideal of $K[\mathcal{A}, X]$.*

The following lemma is a simple method for determining Gröbner bases under a certain condition.

**Lemma C.6** ([8], Lemma 5.66 and Theorem 5.68)**.** *Let $\succ$ be a term ordering and $G = \{g_1, \ldots, g_s\}$. If $\mathtt{LM}(g_i)$ and $\mathtt{LM}(g_j)$ are disjoint for any $i \neq j \in \{1, \ldots, r\}$, then $G$ is a Gröbner basis of $\langle G \rangle$ with respect to $\succ$.*

Recall that $\mathcal{I} : h = \{f \in K[\mathcal{A}, X] \mid fh \in \mathcal{I}\}$ is the ideal quotient of an ideal $\mathcal{I}$ with respect to a polynomial $h$ of $K[\mathcal{A}, X]$. The ideal quotient can be used to break the ideal $\mathcal{I}$ into two ideals as follows.

**Lemma C.7** ([21], Lemma 3.3.6 (Splitting tool))**.** *Let $\mathcal{I}$ be an ideal of $K[\mathcal{A}, X]$ and $h$ a polynomial of $K[\mathcal{A}, X]$. If $\mathcal{I} : h = \mathcal{I} : h^2$, then $\mathcal{I} = (\mathcal{I} : h) \cap (\mathcal{I} + \langle h \rangle)$.*

Lemma C.7 is often used in conjunction with the following lemma, i.e., $\mathcal{I} = (\mathcal{I} : h^m) \cap (\mathcal{I} + \langle h^m \rangle)$ for a sufficiently large integer $m$.

**Lemma C.8** ([21], Proposition 4.3.1 (2))**.** *Let $\mathcal{I}$ be an ideal of $K[\mathcal{A}, X]$ and $U$ a maximal independent set mod $\mathcal{I}$, that is, $U$ is an independent set mod $\mathcal{I}$ with $|U| = \dim(\mathcal{I})$. Let $S = \{\tau_1, \ldots, \tau_l\} \subset \mathcal{I} \subset K[\mathcal{A}, X]$ be a Gröbner basis of $\mathcal{I}K(U)[(\mathcal{A} \cup X) \setminus U]$ and let $h = \mathrm{lcm}(\mathtt{LC}_\succ(\tau_1), \ldots, \mathtt{LC}_\succ(\tau_l)) \in K[U]$. Then, $\mathcal{I}K(U)[(\mathcal{A} \cup X) \setminus U] \cap K[\mathcal{A}, X] = \mathcal{I} : h^m$ for a sufficiently large integer $m$.*

Let $\mathcal{PT}_D$ be the set of all terms of $K[\mathcal{A}]$. Then, $\mathcal{PT}_D \times \mathcal{T}_n := \{\mathcal{A}^\alpha X^\beta \mid \mathcal{A}^\alpha \in \mathcal{PT}_D, X^\beta \in \mathcal{T}_n\}$ is the set of all terms of $K[\mathcal{A}, X]$. For a term ordering $\succ_X$ on $\mathcal{T}_n$ and $\succ_\mathcal{A}$ on $\mathcal{PT}_D$, the product ordering $\succ_{X \times \mathcal{A}}$ is the ordering such that $\mathcal{A}^{\alpha_1} X^{\beta_1} \succ_{X \times \mathcal{A}} \mathcal{A}^{\alpha_2} X^{\beta_2}$ if $X^{\beta_1} \succ_X X^{\beta_2}$ or "$X^{\beta_1} = X^{\beta_2}$ and $\mathcal{A}^{\alpha_1} \succ_\mathcal{A} \mathcal{A}^{\alpha_2}$". In contrast, the product ordering $\succ_{\mathcal{A} \times X}$ is the ordering such that $\mathcal{A}^{\alpha_1} X^{\beta_1} \succ_{\mathcal{A} \times X} \mathcal{A}^{\alpha_2} X^{\beta_2}$ if $\mathcal{A}^{\alpha_1} \succ_\mathcal{A} \mathcal{A}^{\alpha_2}$ or "$\mathcal{A}^{\alpha_1} = \mathcal{A}^{\alpha_2}$ and $X^{\beta_1} \succ_X X^{\beta_2}$". For a product ordering $\succ_{Y_1 \times Y_2}$, we denote by $\mathtt{LC}_{Y_2}(f) \in K[Y_2]$ the leading coefficient of $f$ in $K(Y_2)[Y_1]$ with respect to $\succ_{Y_1}$ for $f \in K[Y_1, Y_2]$. We also say that $\succ_{Y_1 \times Y_2}$ is a block ordering with $Y_1 \succ\succ Y_2$. Let $V(J) = \{q \in K^D \mid f(q) = 0, \forall f \in J\}$ be the variety of an ideal $J$ of $K[\mathcal{A}]$. The following lemma is useful when considering Gröbner bases for specialized parametric ideals.

**Lemma C.9** ([60], Lemma 2.2)**.** *Let $G$ be a Gröbner basis of an ideal $\langle F \rangle$ in $K[\mathcal{A}, X]$ with respect to a product ordering $\succ_{X \times \mathcal{A}}$. If $\sigma_q(\mathtt{LC}_\mathcal{A}(g)) \neq 0$ for each $g \in G \setminus K[\mathcal{A}]$, then for any $q \in V(\langle G \cap K[\mathcal{A}] \rangle) \subset K^D$, $\sigma_q(G)$ is a Gröbner basis of $\langle \sigma_q(F) \rangle$ in $K[X]$ with respect to $\succ_X$.*

We recall that the codimension of an ideal $\mathcal{I}$ of $K[Y]$, denoted by $\mathrm{codim}(\mathcal{I})$, is equal to $|Y| - \dim(\mathcal{I})$. The following lemma can be used to check the radicalness of an unmixed ideal over a field of characteristic 0.

**Lemma C.10** ([61] Proposition 3.65)**.** *Let $K$ be a field of characteristic 0 and $\mathcal{I} = \langle g_1, \ldots, g_s \rangle$ an unmixed ideal of $K[\mathcal{A}, X]$, i.e., all the primary components of a minimal primary decomposition of $\mathcal{I}$*

*have the same codimension. Let $c$ be the codimension of $\mathcal{I}$ and $Jac(g_1, \ldots, g_s) = \left(\frac{\partial(g_1, \ldots, g_s)}{\partial(X, \mathcal{A})}\right)$ the Jacobian matrix of $\{g_1, \ldots, g_s\}$ with respect to $X \cup \mathcal{A}$. Then, the following conditions are equivalent.*

1. *$\mathcal{I}$ is radical,*

2. *there exists a $c \times c$ minor determinant $f$ of $Jac(g_1, \ldots, g_s)$ such that $\mathcal{I} : f = \mathcal{I}$.*

The parametric ideal $\langle \overline{A}G \rangle = \langle f_1, \ldots, f_r \rangle$ satisfies the following properties.

**Proposition C.11.** *Let $\langle \overline{A}G \rangle^e = \langle \overline{A}G \rangle_{K(X)[\mathcal{A}]}$ and $\langle \overline{A}G \rangle^{ec} = \langle \overline{A}G \rangle^e \cap K[\mathcal{A}, X]$. Then,*

1. *$\langle \overline{A}G \rangle^{ec}$ is a prime ideal of $K[\mathcal{A}, X]$,*

2. *$\overline{A}G$ is a Gröbner basis of $\langle \overline{A}G \rangle^e$ with respect to an arbitrary term ordering on $K(X)[\mathcal{A}]$,*

3. *Fix a term ordering $\succ$ on $K(X)[\mathcal{A}]$. For $h = \mathrm{lcm}(\mathtt{LC}_\succ(f_1), \ldots, \mathtt{LC}_\succ(f_r)) \in K[X]$ and a sufficiently large integer $m$*
$$\langle \overline{A}G \rangle^{ec} = \langle \overline{A}G \rangle : h^m,$$

4. *$\mathrm{codim}(\langle \overline{A}G \rangle) \le n$ and $\mathrm{codim}(\langle \overline{A}G \rangle^{ec}) = r$. If $r \ge n$, then $\langle \overline{A}G \rangle^{ec} \not\subset \langle G \rangle_{K[\mathcal{A}, X]}$.*

*Proof.*     1. Since $\overline{A}G$ is a set of linear polynomials over $K(X)$, $\langle \overline{A}G \rangle^e$ is a prime ideal of $K(X)[\mathcal{A}]$. By Lemma C.5, $\langle \overline{A}G \rangle^e \cap K[\mathcal{A}, X]$ is a prime ideal of $K[\mathcal{A}, X]$.

2. Since $f_i$ and $f_j$ do not have common variables except $X$, $\mathtt{LM}(f_i)$ and $\mathtt{LM}(f_j)$ are disjoint for any $i \neq j \in \{1, \ldots, r\}$ with respect to any term ordering on $K(X)[\mathcal{A}]$. Thus, by Lemma C.6, $\overline{A}G$ is a Gröbner basis of $\langle \overline{A}G \rangle^e$ with respect to an arbitrary term ordering on $K(X)[\mathcal{A}]$

3. Since $X$ is a maximal independent set mod $\langle \overline{A}G \rangle$ and $\langle \overline{A}G \rangle^e = \langle \overline{A}G \rangle_{K(X)[\mathcal{A}]} = \langle \overline{A}G \rangle K(X)[\mathcal{A}]$, we obtain the equation from Lemma C.8.

4. Since $\langle G \rangle_{K[X]}$ is a zero-dimensional ideal of $K[X]$,
$$\mathrm{codim}(\langle G \rangle_{K[\mathcal{A}, X]}) = \mathrm{codim}(\langle G \rangle_{K[X]}) = n.$$
As $\langle \overline{A}G \rangle_{K[\mathcal{A}, X]} \subset \langle G \rangle_{K[\mathcal{A}, X]}$, $\mathrm{codim}(\langle \overline{A}G \rangle_{K[\mathcal{A}, X]}) \le \mathrm{codim}(\langle G \rangle)_{K[\mathcal{A}, X]} = n$. Since $\mathrm{codim}(\langle \overline{A}G \rangle^{ec}) = \mathrm{codim}(\langle \overline{A}G \rangle^e)$ and $\overline{A}G$ consists of independent $r$-linear polynomials over $K(X)[\mathcal{A}]$, $\mathrm{codim}(\langle \overline{A}G \rangle^{ec}) = \mathrm{codim}(\langle \overline{A}G \rangle^e) = r$. Assume $r \ge n$. Then $\mathrm{codim}(\langle \overline{A}G \rangle^{ec}) \ge \mathrm{codim}(\langle G \rangle_{K[\mathcal{A}, X]})$. If $\langle \overline{A}G \rangle^{ec} \subset \langle G \rangle_{K[\mathcal{A}, X]}$ then $\langle \overline{A}G \rangle^{ec} = \langle G \rangle_{K[\mathcal{A}, X]}$ since $\langle \overline{A}G \rangle^{ec}$ is a prime ideal by (1). However, this implies $\langle \overline{A}G \rangle^e$ contains a unit $g_1 \in G$ in $K(X)$ and thus $\langle \overline{A}G \rangle^e = K(\mathcal{A})[X]$, which contradicts (1). Therefore, $\langle \overline{A}G \rangle^{ec} \not\subset \langle G \rangle_{K[\mathcal{A}, X]}$.

$\square$

The parametric ideal $\langle \overline{A}G \rangle$ can be decomposed into two ideals as follows.

**Proposition C.12.** *If $r \ge n$, then there exists an ideal $J$ of $K[\mathcal{A}, X]$ such that*
$$\langle \overline{A}G \rangle = \langle G \rangle \cap J$$
*and $\mathrm{codim}(J) \ge \min(r, n+1)$. Moreover, if $r > n$, then $J \cap K[\mathcal{A}] \neq \{0\}$.*

*Proof.* Let $\mathcal{P} = \langle \overline{A}G \rangle^{ec} = \langle \overline{A}G \rangle_{K(X)[\mathcal{A}]} \cap K[\mathcal{A}, X]$. Consider the parameter $a_{0,ij}$ with respect to $\alpha = 0$ of $a_{\alpha, ij} X^\alpha$ in $h_{ij}$. For $i \in \{1, \ldots, s\}$, let $\succ_i$ be a block ordering $\{a_{0,1i}, a_{0,2i}, \ldots, a_{0,ri}\} \succ_i \succ_i \mathcal{A} \setminus \{a_{0,1i}, a_{0,2i}, \ldots, a_{0,ri}\}$ on $K(X)[\mathcal{A}]$. For each $j \in \{1, \ldots, r\}$, the leading term of $f_j = \sum_{k=1}^{s} h_{jk} g_k$ is $g_i a_{0,ji}$ in $K(X)[\mathcal{A}]$ with respect to $\succ_i$ since $g_i a_{0,ji}$ is the only term of $f_j$ that includes $a_{0,ji}$. Thus, $\mathtt{LC}_{\succ_i}(f_j) = g_i$ with respect to $\succ_i$ for each $j$ and $\mathrm{lcm}(\mathtt{LC}_{\succ_i}(f_1), \ldots, \mathtt{LC}_{\succ_i}(f_r)) = \mathrm{lcm}(g_i, \ldots, g_i) = g_i$. By Proposition C.11 (3) and Lemma C.7, $\mathcal{P}$ is a prime ideal of $K[\mathcal{A}, X]$ and, for each $i$,
$$\mathcal{P} = \langle \overline{A}G \rangle : g_i^m \text{ and } \langle \overline{A}G \rangle = \mathcal{P} \cap (\langle \overline{A}G \rangle + \langle g_i^m \rangle)$$

for a sufficiently large integer $m$. Here, $\mathcal{P} \neq \sqrt{(\langle \overline{A}G \rangle)}$ since otherwise $\mathcal{P} = \sqrt{\langle \overline{A}G \rangle} \subset \sqrt{\langle G \rangle} = \langle G \rangle$ (as $\langle G \rangle$ is radical), which contradicts Proposition C.11 (4). Thus, $\langle \overline{A}G \rangle + \langle g_i^m \rangle$ contains at least one isolated component of $\langle \overline{A}G \rangle$. Let $\langle \overline{A}G \rangle + \langle g_1^m \rangle = Q_1 \cap \cdots \cap Q_l$ be a minimal primary decomposition of $\langle \overline{A}G \rangle + \langle g_1^m \rangle$. Without loss of generality, we may assume that $Q_1, \ldots, Q_k$ are isolated primary components of $\langle \overline{A}G \rangle$ and $Q_{k+1}, \ldots, Q_r$ are not for some $k \geq 1$. Since isolated primary components are uniquely determined by Remark C.2, $Q_1, \ldots, Q_k$ are also isolated primary components of $(\langle \overline{A}G \rangle + \langle g_2^m \rangle), \ldots, (\langle \overline{A}G \rangle + \langle g_s^m \rangle)$. As $Q_1 \cap \cdots \cap Q_k \supset \langle \overline{A}G \rangle + \langle g_i^m \rangle$ for each $i$, $Q_1 \cap \cdots \cap Q_k \supset \langle G \rangle^{sm}$. Since $\langle \overline{A}G \rangle = \mathcal{P} \cap Q_1 \cap \cdots \cap Q_l \subset \langle G \rangle$ and $\mathcal{P} \cap Q_{k+1} \cap \cdots \cap Q_l \not\subset \langle G \rangle$, it follows that $Q_1 \cap \cdots \cap Q_k \subset \langle G \rangle$. Thus, $\langle G \rangle = \sqrt{\langle G \rangle} = \sqrt{\langle G \rangle^{sm}} \subset \sqrt{Q_1 \cap \cdots \cap Q_k} = \sqrt{Q_1} \cap \cdots \cap \sqrt{Q_k} \subset \langle G \rangle$, that is, $\langle G \rangle = \sqrt{Q_1} \cap \cdots \cap \sqrt{Q_k}$ is a minimal primary decomposition of $\langle G \rangle$. Let $H = Q_1 \cap \cdots \cap Q_k$ and show that $H$ is radical. As $\langle G \rangle_{K[X]}$ is a zero-dimensional radical ideal of $K[X]$, $\langle G \rangle$ is a $n$-codimensional unmixed radical ideal of $K[\mathcal{A}, X]$. By Lemma C.10, there exists a $n \times n$ minor determinant $f$ of $Jac(g_1, \ldots, g_s)$ such that $\langle G \rangle : f = \langle G \rangle$. Since $\langle \overline{A}G \rangle \subset H$, there exists a generating set $G_H$ of $H$ such that $G_H = \{f_1, \ldots, f_s, \tau_1, \ldots, \tau_u\}$ for some $\tau_1, \ldots, \tau_u \in H$. Then, $Jac(G_H) = \begin{pmatrix} Jac(\overline{A}G) \\ B \end{pmatrix}$ for some $B \in K[\mathcal{A}, X]^{u \times (n+D)}$. As $\sigma_q(\overline{A}G) \supset G$ for some $q \in K^D$, there exists an $n \times n$ minor determinant $g$ of $Jac(G_H)$ such that $\sigma_q(g) = f$. If $H : g \neq H$, then $g \in \sqrt{Q_i}$ for some $i \in \{1, \ldots, k\}$. However, this implies $\sigma_q(g) = f \in \sigma_q(\sqrt{Q_i}) = \sqrt{Q_i}$ and $\langle G \rangle : f = Q_1 \cap \cdots Q_{i-1} \cap Q_{i+1} \cap \cdots \cap Q_k \neq \langle G \rangle$, which contradicts $\langle G \rangle : f = \langle G \rangle$. Therefore, $H : g = H$. Since $H$ is an $n$-codimensional unmixed ideal of $K[\mathcal{A}, X]$, $H$ is radical by Lemma C.10. Then, $H = \sqrt{H} = \sqrt{Q_1} \cap \cdots \cap \sqrt{Q_k} = \langle G \rangle$. Letting $J = \mathcal{P} \cap Q_{k+1} \cap \cdots \cap Q_l$, it follows that $\langle \overline{A}G \rangle = \langle G \rangle \cap J$. By Proposition C.11 (4), $\mathrm{codim}(\mathcal{P}) = r \geq n$. Since $Q_{k+1}, \ldots, Q_l$ are not isolated primary components of $\langle \overline{A}G \rangle$, $\mathrm{codim}(Q_{k+1}), \ldots, \mathrm{codim}(Q_r) \geq n + 1$. Therefore, the codimension of $J$ is $\min(\mathrm{codim}(\mathcal{P}), \mathrm{codim}(Q_{k+1}), \ldots, \mathrm{codim}(Q_l)) \geq \min(r, n + 1)$.

If $r > n$, then $\min(r, n + 1) = n + 1$ and $\dim(J) < (D + n) - (n + 1) = D - 1$. Thus, $\mathcal{A}$ is not an independent set of $J$ since $|\mathcal{A}| = D > D - 1 = \dim(J)$ by Remark C.4, that is, $J \cap K[\mathcal{A}] \neq \{0\}$. $\quad\square$

Recall that $V(J) = \{q \in K^D \mid f(q) = 0, \forall f \in J\}$ is the variety of an ideal $J$ of $K[\mathcal{A}]$. If $J$ is a nonzero ideal, $V(J)$ is a variety of dimension $D - 1$ at most and thus $K^D \setminus V(J)$ is a dense set of $K^D$. Finally, we obtain the proof of Theorem 4.5 as follows.

*proof of Theorem 4.5.* In case $r < n$, $\langle AG \rangle \neq \langle G \rangle$ since $\mathrm{codim}(\langle AG \rangle) = r < n = \mathrm{codim}(\langle G \rangle)$ for any $A \in K[X]_{\leq d}^{r \times s}$. Thus, we assume that $r \geq n$. By Proposition C.12, there exists an ideal $J$ of $K[\mathcal{A}, X]$ such that

$$\langle \overline{A}G \rangle = \langle G \rangle \cap J.$$

1. Assume that $r = n$ and $G$ is a Gröbner basis with respect to a term ordering $\succ_X$ on $K[X]$. Then, for each $i \in \{1, \ldots, n\}$, there exists $g_{k_i} \in G$ such that $\mathrm{LC}_{\succ_X}(g_{k_i}) = x_i^{d_i}$ for some positive integer $d_i$. For simplicity, we may assume that $k_1 = 1, \ldots, k_n = n$. Let $u_i = (0, \ldots, 1, \ldots, 0)$ be the $i$-th unit vector in $K^n$. Consider the parameter $a_{u_i, ii}$ with respect to $X^{u_i} = x_i$ in $h_{ii}$ for each $i$. Fix a term ordering $\succ_{\mathcal{A}}$ such that $\{a_{u_1, 11}, \ldots, a_{u_n, nn}\} \succ \succ \mathcal{A} \setminus \{a_{u_1, 11}, \ldots, a_{u_n, nn}\}$ on $K[\mathcal{A}]$. Let $\succ$ be the product ordering $\succ_{\mathcal{A} \times X}$. Then, $\mathrm{LT}_{\succ}(h_{ii}) = \mathrm{LT}_{\succ}(a_{u_i, ii} x_i g_i) = a_{u_i, ii} x_i^{d_i + 1}$ for each $i \in \{1, \ldots, n\}$. Since $\mathrm{LT}_{\succ}(h_{11}), \ldots, \mathrm{LT}_{\succ}(h_{nn})$ are disjoint, $\overline{A}G$ is a Gröbner basis of $\langle \overline{A}G \rangle$ with respect to $\succ$ by Lemma C.6. For any non-zero polynomial $w(\mathcal{A}) \in K[\mathcal{A}]$, $w(\mathcal{A})g_1 \notin \langle \overline{A}G \rangle$ as $\mathrm{LT}_{\succ}(w(\mathcal{A})g_1) = \mathrm{LT}_{\succ}(w(\mathcal{A}))\mathrm{LT}_{\succ}(g_1) = \mathrm{LT}_{\succ}(w(\mathcal{A}))x_1^{d_1}$ is not divided by any $\mathrm{LT}(h_{11}), \ldots, \mathrm{LT}(h_{nn})$. Thus, such non-zero polynomial $w(\mathcal{A})$ is not in $J$; otherwise $w(\mathcal{A})g_1 \in J \cdot \langle G \rangle \subset \langle G \rangle \cap J = \langle \overline{A}G \rangle$. Hence, $J \cap K[\mathcal{A}] = (\mathcal{P} \cap Q_{k+1} \cap \cdots \cap Q_l) \cap K[\mathcal{A}] = \{0\}$, where $\mathcal{P}, Q_{k+1}, \ldots, Q_l$ are ideals in the proof Proposition C.12. Since $\mathrm{codim}(Q_{k+1}), \ldots, \mathrm{codim}(Q_l) > n$, $Q_i \cap K[\mathcal{A}] \neq \{0\}$ for each $i \in \{k+1, \ldots, l\}$. Therefore, $(\mathcal{P} \cap Q_{k+1} \cap \cdots \cap Q_l) \cap K[\mathcal{A}] = \{0\}$ implies $\mathcal{P} \cap K[\mathcal{A}] = \{0\}$. Let $G' = \{g_1', \ldots, g_l'\}$ be the reduced Gröbner basis of $\mathcal{P}$ with respect to a block ordering $X \succ' \succ' \mathcal{A}$. Obviously, $G' \cap K[\mathcal{A}] = \{0\}$. Then, letting $H = \mathrm{lcm}(\mathrm{LC}_{\mathcal{A}}(g_1) \cdots \mathrm{LC}_{\mathcal{A}}(g_l)) \neq 0$, $\sigma_q(G')$ is a Gröbner basis of $\sigma_q(\mathcal{P})$ for any $q \in V(G' \cap K[\mathcal{A}]) \setminus V(H) = K^D \setminus V(H)$ by Lemma C.9. Fix

$q \in K^D \setminus V(H)$. Since $\langle \overline{A}G \rangle \subset \mathcal{P}$,

$$\langle \sigma_q(\overline{A})G \rangle \subset \sigma_q(\mathcal{P}) \neq K[X].$$

As $\mathcal{P} \not\supset \langle G \rangle_{K[X]}$, there exists $g \in \langle G \rangle_{K[X]} \setminus \mathcal{P}$ such that $\mathrm{LT}_{\succ'}(g) \notin \mathrm{LT}_{\succ'}(\langle G' \rangle)$. This implies $\mathrm{LT}_{\succ'}(g) = \mathrm{LT}_{\succ'}(\sigma_q(g)) \notin \mathrm{LT}_{\succ'}(\langle \sigma_q(G') \rangle) = \mathrm{LT}_{\succ'}(\sigma_q(\mathcal{P}))$ since $g = \sigma_q(g)$ and $\sigma_q(G')$ is a Gröbner basis of $\sigma_q(\mathcal{P})$ with respect to $\succ'$. Thus, $g \in \langle G \rangle \setminus \sigma_q(\mathcal{P})$ and $\langle G \rangle \not\subset \sigma_q(\mathcal{P})$. As $\langle \sigma_q(\overline{A})G \rangle \subset \sigma_q(\mathcal{P})$, we obtain $\langle \sigma_q(\overline{A})G \rangle \neq \langle G \rangle$. With $H \neq 0$, $K^D \setminus V(H)$ is a dense set of $K^D$.

2. Assume that $r > n$. Then we can take $J$ with $J \cap K[\mathcal{A}] \neq \{0\}$ by Proposition C.12. Fix $q \in K^D \setminus V(J \cap K[\mathcal{A}])$. Then, for $0 \neq f(\mathcal{A}) \in J \cap K[\mathcal{A}]$, it follows that $0 \neq f(q) \in \sigma_q(J)$, that is, $\sigma_q(J) = K[X]$. Since $\langle G \rangle \cdot J \subset \langle G \rangle \cap J = \langle \overline{A}G \rangle$ and $\sigma_q(\langle G \rangle) = \langle G \rangle$,

$$\langle \sigma_q(\overline{A})G \rangle = \sigma_q(\langle \overline{A}G \rangle) \supset \sigma_q(\langle G \rangle \cdot J) = \sigma_q(\langle G \rangle) \cdot \sigma_q(J) = \langle G \rangle.$$

Since $\langle \sigma_q(\overline{A})G \rangle \subset \langle G \rangle$ always holds, we obtain $\langle \sigma_q(\overline{A})G \rangle = \langle G \rangle$. As $J \cap K[\mathcal{A}] \neq 0$, $K^D \setminus V(J \cap K[\mathcal{A}])$ is a dense set of $K^D$.

$\square$

**Corollary 4.6.** *Let $d$ and $d_{\max}$ be positive integers. Let $K = \mathbb{F}_p$ be a finite field of order $p$ for a prime number $p$ and let $G$ be a subset of $\mathbb{F}_p[X]_{\leq d_{\max}}$ such that $\langle G \rangle$ is a 0-dimensional radical ideal of $\mathbb{F}_p[X]$. Assume $r > n$ and let $\mathcal{G} = \{A \in \mathbb{F}_p[X]_{\leq d}^{r \times s} \mid \langle AG \rangle = \langle G \rangle\}$. Then, a generically sampled $A \in \mathbb{F}_p[X]_{\leq d}^{r \times s}$ satisfies $\langle AG \rangle = \langle G \rangle$ with probability*

$$\Pr(p) = \frac{|\mathcal{G}|}{|\mathbb{F}_p[X]_{\leq d}^{r \times s}|} \geq 1 - \frac{d'}{p} \tag{4.4}$$

*for some positive integer $d'$, which is determined by $d$ and $d_{\max}$, independent of any specific $p$.*

*proof of Corollary 4.6.* Without loss of generality, we may assume that $G = \{g_1, \ldots, g_s\} \subset \mathbb{Z}[X]$, $\overline{A} \in \mathbb{Z}[\mathcal{A}, X]^{r \times s}$, $\overline{A}G = \{f_1, \ldots, f_r\} \subset \mathbb{Z}[\mathcal{A}, X]$, and $\langle G \rangle_{\mathbb{Q}[\mathcal{A}, X]}$ is a 0-dimensional radical ideal of $\mathbb{Q}[\mathcal{A}, X]$. By Proposition C.12, there exists an ideal $J$ of $\mathbb{Q}[\mathcal{A}, X]$ such that $\langle \overline{A}G \rangle_{\mathbb{Q}[\mathcal{A}, X]} = \langle G \rangle_{\mathbb{Q}[\mathcal{A}, X]} \cap J$ and $J \cap \mathbb{Q}[\mathcal{A}] \neq \{0\}$. For a subset $L$ of $\mathbb{Z}[\mathcal{A}, X]$, we simply write $\langle L \rangle_{\mathbb{Q}} = \langle L \rangle_{\mathbb{Q}[\mathcal{A}, X]}$ and $\langle L \rangle_p = \langle \phi_p(L) \rangle_{\mathbb{F}_p[\mathcal{A}, X]}$ respectively, where $\phi_p : \mathbb{Z}[\mathcal{A}, X] \to \mathbb{F}_p[\mathcal{A}, X]$ is the canonical projection. Let $\mathcal{P}, Q_{k+1}, \ldots, Q_l$ be ideals in the proof of Proposition C.12, that is, $J = \mathcal{P} \cap Q_{k+1} \cap \cdots \cap Q_l$. Then

$$
\begin{aligned}
\langle \overline{A}G \rangle_{\mathbb{Q}} : \langle G \rangle_{\mathbb{Q}} &= (\langle G \rangle_{\mathbb{Q}} \cap J) : \langle G \rangle_{\mathbb{Q}} = (\langle G \rangle_{\mathbb{Q}} \cap (\mathcal{P} \cap Q_{k+1} \cdots \cap Q_l)) : \langle G \rangle_{\mathbb{Q}} \\
&= (\langle G \rangle_{\mathbb{Q}} : \langle G \rangle_{\mathbb{Q}}) \cap (\mathcal{P} : \langle G \rangle_{\mathbb{Q}}) \cap (Q_{k+1} : \langle G \rangle_{\mathbb{Q}}) \cap \cdots \cap (Q_l : \langle G \rangle_{\mathbb{Q}}) \\
&= \mathcal{P} \cap (Q_{k+1} : \langle G \rangle_{\mathbb{Q}}) \cap \cdots \cap (Q_l : \langle G \rangle_{\mathbb{Q}}).
\end{aligned}
$$

Fix a block ordering $X \succ\succ \mathcal{A}$. For new variables $T = \{t_1, \ldots, t_{s-1}\}$ and $y$, let $\omega = g_1 + g_2 t_1 + \cdots g_s t_{s-1} \in \mathbb{Q}[T, \mathcal{A}, X]$, $S = \{yf_1, \ldots, yf_r, (1-y)\omega\} \subset \mathbb{Q}[y, T, \mathcal{A}, X]$, and $G_S$ a Gröbner basis of $\langle S \rangle_{\mathbb{Q}[y, T, \mathcal{A}, X]}$ with respect to an extended block ordering $\{y\} \succ\succ T \succ\succ X \succ\succ \mathcal{A}$. It is known that $G' = ((G_S \cap \mathbb{Q}[T, \mathcal{A}, X]) \cdot \omega^{-1}) \cap \mathbb{Q}[\mathcal{A}, X]$ is a Gröbner basis of $\langle \overline{A}G \rangle_{\mathbb{Q}} : \langle G \rangle_{\mathbb{Q}}$ with respect to $\succ$ (see Lemma 1.8.12 in [21]). Since an upper bound of degrees of reduced Gröbner bases can be decided from degrees of the generator, there exists $d'$ such that $d' > \deg(g)$ for any $g \in G'$ derived from any $G \in \mathbb{Z}[\mathcal{A}, X]_{\leq d_{\max}}$ and $A \in \mathbb{Z}[X]_{\leq d}^{r \times s}$. In other words, $d'$ is determined by $d$ and $d_{\max}$, independent of any specific $p$. Let $f \in (\langle \overline{A}G \rangle_{\mathbb{Q}} : \langle G \rangle_{\mathbb{Q}}) \cap (\mathbb{Z}[\mathcal{A}] \setminus p\mathbb{Z}[\mathcal{A}])$ then $\deg f < d'$ and $\phi_p(f) \neq 0$. Since $\{Q_1, \ldots, Q_l\}$ is a minimal primary decomposition of $\langle \overline{A}G \rangle_{\mathbb{Q}} + \langle g_1^m \rangle_{\mathbb{Q}}$, for each $i \in \{k+1, \ldots, l\}$, $\langle G \rangle_{\mathbb{Q}} = Q_1 \cap \cdots \cap Q_k \not\subset Q_i$ and thus $Q_i : \langle G \rangle_{\mathbb{Q}}$ is a primary ideal with $\sqrt{Q_i : \langle G \rangle_{\mathbb{Q}}} = \sqrt{Q_i}$. Hence, $\sqrt{\langle \overline{A}G \rangle_{\mathbb{Q}} : \langle G \rangle_{\mathbb{Q}}} = \mathcal{P} \cap \sqrt{(Q_{k+1} : \langle G \rangle_{\mathbb{Q}})} \cap \cdots \cap \sqrt{(Q_l : \langle G \rangle_{\mathbb{Q}})} = \mathcal{P} \cap \sqrt{Q_{k+1}} \cap \cdots \cap \sqrt{Q_l} = \sqrt{J}$ and $f \in \sqrt{J}$, i.e., $f^M \in J \cap \mathbb{Z}[\mathcal{A}]$ for some positive integer $M$. Let $f_p = \phi_p(f) \in \mathbb{F}_p[\mathcal{A}]$ and $J_{\mathbb{Z}} = J \cap \mathbb{Z}[\mathcal{A}, X]$, then $f_p^M \in \phi_p(J_{\mathbb{Z}} \cap \mathbb{Z}[\mathcal{A}])$. Here, $\langle \overline{A}G \rangle_p \supset \langle G \rangle_p \cdot \phi_p(J_{\mathbb{Z}})$ since $\langle \overline{A}G \rangle_{\mathbb{Q}} \supset \langle G \rangle_{\mathbb{Q}} \cdot J \supset \langle G \rangle_{\mathbb{Q}} \cdot J_{\mathbb{Z}}$. For $q \in \mathbb{F}_p^D$ with $f_p(q) \neq 0$, $\sigma_q(\phi_p(J_{\mathbb{Z}})) = \mathbb{F}_p[X]$ as $0 \neq f_p^M(q) \in \sigma_q(\phi_p(J_{\mathbb{Z}}))$, and

thus $\sigma_q(\langle \overline{A}G \rangle_p) \supset \sigma_q(\langle G \rangle_p \cdot \phi_p(J_{\mathbb{Z}})) = \sigma_q(\langle G \rangle_p) \cdot \sigma_q(\phi_p(J_{\mathbb{Z}})) = \langle G \rangle_{\mathbb{F}_p[X]} \cdot \mathbb{F}_p[X] = \langle G \rangle_{\mathbb{F}_p[X]}$.
As the inverse inclusion is obvious, we obtain $\sigma_q(\langle \overline{A}G \rangle_p) = \langle G \rangle_{\mathbb{F}_p[X]}$ for any $q \in \mathbb{F}_p^D \setminus V(f_p)$.
Since $f_p$ has at most $\deg(f_p)p^{D-1} \leq \deg(f)p^{D-1} \leq d'p^{D-1}$ solutions in $\mathbb{F}_p^D$ (see Theorem 6.13 in
[45]), $\Pr(p) \geq \frac{|\mathbb{F}_p^D \setminus V(f_p)|}{|\mathbb{F}_p^D|} \geq \frac{p^D - d'p^{D-1}}{p^D} = 1 - \frac{d'}{p}$ and this goes 1 for a sufficiently large $p$. $\qquad\square$

## C.2  Empirical results

We conduct a numerical experiment to justify our backward transform based on Theorem 4.5. The
border basis generation and sampling $A$ for backward transform follows the main experiment of
training the Transformer oracle. See Section 5.2. Figure 6 shows the success rate of having $F = AG$
such that $\langle F \rangle = \langle G \rangle$. As Theorem 4.5 and Corollary 4.6 suggest, the success rate is zero for $|F| = n$
and for $|F| > n$ improves with larger $n$, $p$, and $|F|$.

## D  Monomial embedding

### D.1  Implementation details

We here elaborate the implementation details of monomial embedding method. First, recall the
definition.

**Definition 4.7.** (Monomial embedding) Let $\Sigma$ be the set of all tokens. Let $(t, \texttt{<*>})$ be a pair consisting
of a monomial $t = cx^{\boldsymbol{a}} \in \mathcal{T}_n$ with coefficient $c \in K$, exponent vector $\boldsymbol{a} \in \mathbb{Z}_{\geq 0}^n$, and a follow-up
token $\texttt{<*>} \in \Sigma$. Let $\varphi_c$, $\varphi_e$, and $\varphi_f$ denote embeddings of the coefficient, exponent vector, and follow-
up token into a $d$-dimensional space, respectively. The monomial embedding $\varphi_m : \mathcal{T}_n \times \Sigma \to \mathbb{R}^d$ is
given by

$$\varphi_m(t, \texttt{<*>}) = \varphi_c(c) + \varphi_e(\boldsymbol{a}) + \varphi_f(\texttt{<*>}). \qquad (4.6)$$

**Embedding maps.**  The embedding maps $\varphi_c$ and $\varphi_f$ are standard token embeddings implemented
using trainable embedding matrices. The former is used for coefficient tokens, while the latter is used
for special tokens such as $\texttt{[SEP]}$ or $\texttt{[PAD]}$.

The embedding map $\varphi_e$ is designed to handle exponent vectors and is conceptually realized using
$n$ independent embedding maps, one for each variable. Specifically, the $i$-th embedding map
$\varphi_e^{(i)}$ processes the exponent token corresponding to the variable $x_i$. Given an exponent vector
$\boldsymbol{a} = (a_1, \ldots, a_n)$, its embedding is computed as

$$\varphi_e(\boldsymbol{a}) = \frac{1}{n} \sum_{i=1}^{n} \varphi_e^{(i)}(a_i). \qquad (D.1)$$

**Unembedding.**  After processing the input embeddings, the Transformer outputs are passed through
an unembedding layer before reaching the classification head. Since our tokenization scheme based on
monomial embeddings reduces the number of tokens to approximately $1/(n+1)$ of that in the standard
representation, the unembedding layer restores the original token structure prior to classification.
Concretely, the unembedding can be implemented by a linear transformation $\psi : \mathbb{R}^{1 \times d} \to \mathbb{R}^{(n+1) \times d}$,
which is applied to each embedding vector to expand it back to a sequence of $(n + 1)$ vectors. This
operation reconstructs the original token alignment, making it compatible with downstream tasks
such as sequence classification or generation.

**The $\texttt{<bos>}$ token.**  The Transformer decoder requires the right-shift operation. To this end, we
append, e.g., $\texttt{<bos>}$, to the input sequence before monomial-tokenized. As a consequence, we have
$n + 1$ redundant tokens; these token can be simply eliminated.

### D.2  Empirical validations of monomial embedding

**Task.**  Given a list of polynomials $f_1, \ldots, f_r \in \mathbb{F}_p[x_1, \ldots, x_n]$, the task is to compute their cumula-
tive products $f_1, f_1 f_2, \ldots, \prod_{i=1}^{r} f_i$.

Table 3: Comparison between baseline (infix tokenization) and proposed (monomial tokenization and embedding) methods across different numbers of variables. The success rate measures the successful generation of complete cumulative products.

| # Variables | Method | Success rate (%) | GPU memory (MB) |
|---|---|---|---|
| $n = 2$ | infix | 33.9 | 4,302 |
| | monomial | 39.7 | 1,678 |
| $n = 3$ | infix | 37.5 | 11,296 |
| | monomial | 44.6 | 2,408 |
| $n = 4$ | infix | 38.8 | 23,632 |
| | monomial | 47.3 | 3,260 |
| $n = 5$ | infix | 22.0 | 27,442 |
| | monomial | 53.7 | 3,424 |

The polynomial product task requires the model to understand addition and product, and thus this is a basic symbolic computation over a ring. Several studies have reported that learning symbolic tasks over finite fields is difficult [35, 63]. Even learning a simple parity function $f : (x_1, \ldots, x_m) \in \{-1, 1\}^m \mapsto \prod_{i=1}^{m} x_i$, which corresponds to a scalar product over $\mathbb{F}_2$, is theoretically known to be hard [57]. However, recent work has shown that auto-regressive generation can overcome this challenge [37], motivating our adoption of a sequential formulation.

**Setup.** The Transformer architecture and training setup follow Section 5.2. Transformer models were trained on 100,000 samples and evaluated on 1,000 samples. For each sample, the number of polynomials $r$ was uniformly sampled from $\{2, 3, 4\}$. Then, polynomials $f_1, \ldots, f_r \in \mathbb{F}_7[x_1, \ldots, x_n]$ were sampled with a maximum degree $d_{\max} = 4$ and a maximum number of terms $t_{\max} = 5$. The sampling follows the strategy given in Appendix E.1, except that the polynomial degree was uniformly sampled from $\mathbb{U}[1, d_{\max}]$ to exclude the zero polynomial and avoid trivially easy product computations. We tested cases with $n = 2, 3, 4, 5$. We adopted the standard infix representation as a baseline (Equation (4.5)). Note that the Transformer model with the proposed monomial embedding predicts target sequences in the same representation via the unembedding layer. Thus, this baseline setup allows us to directly assess the impact of the proposed tokenization and embedding strategy.

**Results.** Figure 7 shows the average number of tokens in test samples for both the baseline method and the proposed method. The shaded region indicates the range from minimum to maximum. A large gap in both the average and maximum number of tokens between the infix and monomial embeddings can be observed. This gap is also reflected in the GPU memory consumption in Table 3. Notably, the memory consumption of the monomial embedding with $n = 5$ is lower than that of the infix embedding with $n = 2$. The monomial embedding is further advantageous for learning, as indicated by its success rate (i.e., the proportion of samples for which a complete sequence of cumulative products is successfully generated). The improvement in success rate becomes even more pronounced for larger values of $n$. Considering the correspondence between monomials is essential in symbolic computation. With infix embedding, the model must establish attention between $(n + 1)$ tokens and another $(n + 1)$ tokens, resulting in increased complexity. In contrast, monomial embedding reduces this to a one-to-one correspondence, which benefits both memory efficiency and success rate.

### D.3 Reduction profile of the input sequences.

Figure 8 completes Figure 2. We analyze the input sequence reduction in the test samples used in our main experiments. As mentioned in Section 4.2, we used a minimal subset $\mathcal{L}' \in \mathcal{L}$ that allows us to retrieve the universe $\mathcal{L}$, which is an order ideal. Particularly, we used the *corner terms* [40], which we define next.

**Definition D.1** (Divisibility of Monomials). Let

$$t = x_1^{a_1} x_2^{a_2} \cdots x_n^{a_n} \quad \text{and} \quad s = x_1^{b_1} x_2^{b_2} \cdots x_n^{b_n}$$

be monomials in the polynomial ring $K[X]$. We say that $t$ *divides* $s$, written $t \mid s$, if and only if

$$a_i \leq b_i \quad \text{for all } i = 1, \ldots, n,$$

equivalently, if there exists a monomial $u$ such that $s = tu$.

**Definition D.2** (Corner Terms). Let $\mathcal{L}$ be an order ideal in $K[X]$. The *corner terms* of $\mathcal{L}$ are its maximal monomials, i.e. those $t \in \mathcal{L}$ such that whenever $t \mid s$ with $s \in \mathcal{L}$, one has $s = t$. Equivalently,

$$\mathcal{L}' = \big\{\, t \in \mathcal{L} \mid (\forall s \in \mathcal{L})\,(\,t \mid s \;\Rightarrow\; s = t\,)\big\}.$$

Geometrically, each $c \in \mathcal{L}'$ is the "corner" of an axis-aligned hyper-box in $\mathbb{N}^n$ whose lattice points are precisely the divisors of $c$, and the union of these boxes is the entire order ideal.

# E   Experiment setup and additional results

## E.1   Sampling polynomials.

Random sampling a polynomial from $\mathbb{F}_p[X]_{\leq d_{\max}}$ with some degree bound $d_{\max}$ is a basic operation in our experiments. Here, *random sampling* can be defined in several ways. The mathematically generic way is uniformly sampling the coefficients of polynomial $f = \sum_{\boldsymbol{a}: \|\boldsymbol{a}\|_1 \leq d_{\max}} c_{\boldsymbol{a}} x^{\boldsymbol{a}}$. It is worth noting that in such as case, $f$ almost always dense and of degree $d_{\max}$.

However, this approach is not always reasonable from the practical scenario. The maximum degree $d_{\max}$ can only mean the limit of the highest acceptable degree, and the user-input polynomial can vary between constant to degree-$d_{\max}$ ones dynamically.

Taking into account this, random sampling in this paper is performed by first uniformly samlping the degree and the number of terms of polynomial. Namely, given the predesgianated maximum degree $d_{\max}$ and the maximum number of terms $t_{\max}$, a polynomial is sampled with $d \sim \mathbb{U}[0, d_{\max}]$ and $t_{\max} \sim \mathbb{U}[0, \bar{t}_{\max}]$, where $\bar{t}_{\max} = \min\left(t_{\max}, \binom{n + d_{\max}}{n}\right)$, and $\mathbb{U}[a, b]$ is a uniform distribution with range $[a, b] \subset \mathbb{Z}$.

## E.2   Comparisons of polynomial systems in literature.

To the best of our knowledge, our experiments cover the largest and the most general class of polynomial systems in the literature of deep-learning based Gröbner/border basis computation [35, 54].

- The experiments in [54] mostly focus on $n = 3$ variables and binomials (for both input systems $F$ and Gröbner bases $G$).
- The experiments in [35] handle up to $n = 5$, but for $n > 1$, the mixing matrix $A$ of $F = AG$ is sparsified to keep the input system size moderate (i.e., to keep the number of tokens $< 5000$). The design of $A$ only supports an input system of size $|F| \geq |G|$. Besides, the ideals are restricted to those in shape position, where $G$ is restricted to size $|G| = n$.

Table 4 compares the scale and class of systems and ideals ($n = 5$, numbers are rounded).

## E.3   Learning successful expansions

**Computational resources.**   Training was performed on a system with 48-core CPUs, 768 GB of RAM, and NVIDIA RTX A6000 Ada GPUs. Each run completed in less than a day on a single GPU.

**Datasets.**   To construct the training set, we first generated one million border bases $\{G_i\}_i$ as described in Section 4.1.1. These were then transformed into an equal number of non-bases $\{F_i\}_i$ using the backward generator transform from Section 4.1.2. Subsequently, the Improved Border Basis

Table 4: Comparison of the scale and class of systems and ideals ($n = 5$).

| Reference | Input system size $|F|$ | Avg. #terms in $F$ | Class of ideals/systems |
|---|---|---|---|
| [54] | 10 | 20 | binomial |
| [35] | 5–7 | 42 | shape position ($|F| \geq |G| = n$) |
| Ours | 6–10 | 130 | vanishing ideal ($|G| \geq n$, $|F| < |G|$ allowed) |

Table 5: Evaluation results of Transformer predictions over polynomial rings $\mathbb{F}_p[x_1, \ldots, x_n]$. Metrics are reported for different values of $l$.

| Field | Variables | $l$ | Precision (%) | Recall (%) | F1 Score (%) | No Expansion Acc. (%) |
|---|---|---|---|---|---|---|
| | | 1 | 79.4 | 81.2 | 80.3 | 96.3 |
| | $n = 3$ | 3 | 84.1 | 86.1 | 85.1 | 96.0 |
| | | 5 | 85.6 | 87.6 | 86.6 | 96.6 |
| | | 1 | 85.1 | 86.6 | 85.8 | 98.8 |
| $\mathbb{F}_7$ | $n = 4$ | 3 | 88.4 | 89.1 | 88.8 | 98.8 |
| | | 5 | 89.8 | 91.1 | 90.4 | 98.8 |
| | | 1 | 91.4 | 91.9 | 91.7 | 99.1 |
| | $n = 5$ | 3 | 93.0 | 93.3 | 93.1 | 99.1 |
| | | 5 | 93.9 | 94.6 | 94.2 | 98.7 |
| | | 1 | 84.4 | 86.8 | 85.6 | 99.7 |
| | $n = 3$ | 3 | 89.4 | 90.0 | 89.7 | 99.7 |
| | | 5 | 91.6 | 93.2 | 92.4 | 99.7 |
| | | 1 | 90.7 | 91.6 | 91.1 | 98.8 |
| $\mathbb{F}_{31}$ | $n = 4$ | 3 | 92.9 | 93.7 | 93.3 | 98.8 |
| | | 5 | 94.2 | 94.7 | 94.4 | 98.8 |
| | | 1 | 92.7 | 93.1 | 92.9 | 99.6 |
| | $n = 5$ | 3 | 94.3 | 94.6 | 94.4 | 99.6 |
| | | 5 | 94.8 | 95.3 | 95.1 | 99.6 |
| | | 1 | 87.8 | 88.3 | 88.1 | 99.7 |
| | $n = 3$ | 3 | 91.3 | 91.8 | 91.5 | 99.7 |
| | | 5 | 93.6 | 93.7 | 93.7 | 99.7 |
| | | 1 | 92.0 | 93.2 | 92.6 | 99.6 |
| $\mathbb{F}_{127}$ | $n = 4$ | 3 | 94.5 | 94.9 | 94.7 | 99.6 |
| | | 5 | 94.9 | 95.5 | 95.2 | 99.6 |
| | | 1 | 93.1 | 94.7 | 93.9 | 99.6 |
| | $n = 5$ | 3 | 94.4 | 95.8 | 95.1 | 99.6 |
| | | 5 | 94.8 | 96.0 | 95.4 | 99.6 |

(IBBA; [29]) algorithm was executed on each $F_i$. From each run, we collected samples only from the last five expansion calls in the final loop (cf. Algorithm 1). This process yielded approximately five million samples, from which we randomly selected one million without replacement to form the training set. The test set of 1,000 samples was constructed in the same manner.

**Additional Results.** Table 5 is the complete version of Table 1. The overall trend is the same.

## E.4   Out of Distribution Experiments

We report our results on the out-of-distribution performance of our approach. In Figure 9 and Figure 10, we compare the number of fallbacks and the achieved speedup, respectively, across different configurations for $n = 3$ variables. The columns correspond to the degree of the transformation matrix $A$ (ranging from 2 to 4), while the rows represent the total degree, increasing from 4 in the top row to 6 in the bottom row. Comprehensive results on the out-of-distribution performance for varying degrees are further provided in Tables 7 to 9.

## E.5   Border Basis Bottleneck

To characterize the computational bottleneck of the border-basis algorithm, we measure the fraction of the total runtime spent in the final stage (i.e., the phase after the last universe enlargement) and, within that stage, the fraction attributable to the last $k \in \{1, 2, 3, 4, 5\}$ Gaussian-elimination steps. Our empirical results in Table 10 indicate that the final stage overwhelmingly dominates execution time—on average consuming about $95\%$ of the total runtime. Moreover, the last five Gaussian eliminations alone account for roughly $70\%$–$95\%$ of the final-stage runtime.

Table 6: Mean $\pm$ standard deviation of the number of zero reductions for IBBA vs. OBBA. Here we see an up to 3.5 improvement in the number of zero reductions for the in distribution setting with total degree 4 and degree of transformation 1 for $n = 5$.

| Field | Total Degree | Degree Transformation | IBBA | OBBA |
|---|---|---|---|---|
| $\mathbb{F}_7$ | 2 | 1 | $1369.31_{\pm\,613.27}$ | $380.15_{\pm\,140.33}$ |
| $\mathbb{F}_7$ | 3 | 1 | $1854.63_{\pm\,1114.23}$ | $638.12_{\pm\,505.61}$ |
| $\mathbb{F}_7$ | 4 | 1 | $3177.67_{\pm\,2930.36}$ | $1746.99_{\pm\,1992.24}$ |
| $\mathbb{F}_{127}$ | 2 | 1 | $1409.02_{\pm\,596.12}$ | $390.36_{\pm\,155.47}$ |
| $\mathbb{F}_{127}$ | 3 | 1 | $2224.25_{\pm\,1478.50}$ | $1025.64_{\pm\,785.64}$ |
| $\mathbb{F}_{127}$ | 4 | 1 | $2510.42_{\pm\,2006.56}$ | $1112.15_{\pm\,1190.61}$ |
| $\mathbb{F}_{31}$ | 2 | 1 | $1324.05_{\pm\,647.51}$ | $361.22_{\pm\,155.87}$ |
| $\mathbb{F}_{31}$ | 3 | 1 | $2010.02_{\pm\,1364.68}$ | $797.14_{\pm\,695.37}$ |
| $\mathbb{F}_{31}$ | 4 | 1 | $3167.66_{\pm\,2592.95}$ | $1672.20_{\pm\,1555.12}$ |

Table 7: Wall-clock runtime in seconds (mean $\pm$ standard deviation over 100 random zero-dimensional systems of total degree $d \leq 4$ and max degree 1 of $A$) for five algorithms over polynomial rings $\mathbb{F}_7$, $\mathbb{F}_{127}$, and $\mathbb{F}_{31}$ in $n = 5$ variables. "Baseline" comprises BBA and IBBA; "Ours" comprises OBBA, IBBA+FGE, and OBBA+FGE. OBBA+FGE is the fastest in every setting.

| Field | $d$ | Baseline | | Ours | | |
|---|---|---|---|---|---|---|
| | | BBA | IBBA | OBBA | IBBA+FGE | OBBA+FGE |
| $\mathbb{F}_7$ | 2 | $10.49 \pm 7.37$ | $7.16 \pm 4.55$ | $2.43 \pm 1.24$ | $0.86 \pm 0.46$ | $\mathbf{0.59} \pm 0.31$ |
| | 3 | $26.45 \pm 34.13$ | $19.61 \pm 27.10$ | $8.30 \pm 12.60$ | $1.88 \pm 2.12$ | $\mathbf{1.30} \pm 1.45$ |
| | 4 | $156.64 \pm 581.66$ | $101.22 \pm 365.21$ | $67.82 \pm 282.15$ | $7.44 \pm 24.82$ | $\mathbf{5.51} \pm 19.41$ |
| $\mathbb{F}_{127}$ | 2 | $13.44 \pm 8.52$ | $9.04 \pm 5.34$ | $3.06 \pm 1.49$ | $1.01 \pm 0.49$ | $\mathbf{0.67} \pm 0.31$ |
| | 3 | $49.90 \pm 74.57$ | $33.86 \pm 47.53$ | $20.97 \pm 34.87$ | $2.75 \pm 3.14$ | $\mathbf{2.09} \pm 2.53$ |
| | 4 | $77.89 \pm 132.87$ | $53.57 \pm 93.82$ | $39.91 \pm 87.16$ | $4.62 \pm 7.55$ | $\mathbf{3.47} \pm 6.23$ |
| $\mathbb{F}_{31}$ | 2 | $11.44 \pm 8.25$ | $7.60 \pm 5.13$ | $2.58 \pm 1.37$ | $0.88 \pm 0.49$ | $\mathbf{0.60} \pm 0.32$ |
| | 3 | $40.65 \pm 76.90$ | $27.36 \pm 47.71$ | $13.87 \pm 26.07$ | $2.38 \pm 3.32$ | $\mathbf{1.66} \pm 2.35$ |
| | 4 | $136.68 \pm 244.07$ | $97.84 \pm 173.36$ | $68.77 \pm 158.72$ | $7.42 \pm 12.01$ | $\mathbf{5.63} \pm 9.34$ |

### E.6 Border Gap vs Border Distance

Detecting the final stage of Border Basis computation is critical for the efficiency of the proposed method. We empirically investigate the relationship between border distance, which is unknown in practice, and border gap, which can be an input parameter as we can measure it. To make the border gap independent of the scale of the problem, we consider the relative border gap defined as $\frac{|\mathcal{V}|}{|\mathcal{L}|}$. Our results are summarised in Figures 11 to 13, indicating that the border gap is a suitable proxy for the border distance.

Table 8: Wall-clock runtime in seconds (mean $\pm$ standard deviation over 100 random zero-dimensional systems of total degree $d \leq 4$ and max degree 1 for each variable) for five algorithms over polynomial rings $\mathbb{F}_{127}$, $\mathbb{F}_{31}$, and $\mathbb{F}_7$ in $n = 4$ variables. "Baseline" comprises BBA and IBBA; "Ours" comprises OBBA, IBBA+FGE, and OBBA+FGE. OBBA+FGE is the fastest in almost every setting.

| Field $d$ | | Baseline | | Ours | | |
| | | BBA | IBBA | OBBA | IBBA+FGE | OBBA+FGE |
| --- | --- | --- | --- | --- | --- | --- |
| $\mathbb{F}_{127}$ | 2 | $0.43 \pm 0.42$ | $0.33 \pm 0.24$ | $0.20 \pm 0.11$ | $0.09 \pm 0.05$ | $\mathbf{0.09} \pm 0.04$ |
| | 3 | $1.71 \pm 2.35$ | $1.29 \pm 1.61$ | $0.62 \pm 0.64$ | $\mathbf{0.23} \pm 0.21$ | $0.24 \pm 0.25$ |
| | 4 | $2.28 \pm 3.66$ | $1.76 \pm 2.56$ | $0.81 \pm 1.00$ | $\mathbf{0.29} \pm 0.33$ | $0.29 \pm 0.33$ |
| $\mathbb{F}_{31}$ | 2 | $0.46 \pm 0.43$ | $0.35 \pm 0.25$ | $0.22 \pm 0.13$ | $0.09 \pm 0.05$ | $\mathbf{0.09} \pm 0.05$ |
| | 3 | $1.39 \pm 1.59$ | $1.07 \pm 1.18$ | $0.54 \pm 0.53$ | $0.21 \pm 0.18$ | $\mathbf{0.21} \pm 0.18$ |
| | 4 | $4.65 \pm 7.77$ | $3.64 \pm 6.10$ | $2.12 \pm 3.61$ | $0.51 \pm 0.66$ | $\mathbf{0.50} \pm 0.59$ |
| $\mathbb{F}_7$ | 2 | $0.36 \pm 0.27$ | $0.29 \pm 0.17$ | $0.19 \pm 0.09$ | $0.08 \pm 0.03$ | $\mathbf{0.08} \pm 0.03$ |
| | 3 | $1.25 \pm 1.68$ | $0.94 \pm 1.19$ | $0.48 \pm 0.57$ | $0.18 \pm 0.17$ | $\mathbf{0.18} \pm 0.18$ |
| | 4 | $4.89 \pm 11.42$ | $4.11 \pm 9.94$ | $2.13 \pm 5.71$ | $0.55 \pm 0.94$ | $\mathbf{0.52} \pm 0.84$ |

Table 9: Wall-clock runtime in seconds (mean $\pm$ standard deviation over 100 random zero-dimensional systems of total degree $4 \leq d \leq 6$ and max degree 1) for five algorithms over polynomial rings $\mathbb{F}_{31}$, $\mathbb{F}_7$, and $\mathbb{F}_{127}$ for $n = 3$ variables. "Baseline" comprises BBA and IBBA; "Ours" comprises OBBA, IBBA+FGE, and OBBA+FGE. OBBA+FGE (last column) is the fastest in every setting.

| Field $d$ | | Baseline | | Ours | | |
| | | BBA | IBBA | OBBA | IBBA+FGE | OBBA+FGE |
| --- | --- | --- | --- | --- | --- | --- |
| $\mathbb{F}_{31}$ | 4 | $0.07 \pm 0.11$ | $0.06 \pm 0.09$ | $0.06 \pm 0.08$ | $0.03 \pm 0.03$ | $\mathbf{0.03} \pm 0.03$ |
| | 5 | $0.30 \pm 0.76$ | $0.24 \pm 0.56$ | $0.19 \pm 0.45$ | $0.07 \pm 0.12$ | $\mathbf{0.07} \pm 0.11$ |
| | 6 | $0.42 \pm 0.77$ | $0.36 \pm 0.69$ | $0.28 \pm 0.53$ | $0.10 \pm 0.15$ | $\mathbf{0.10} \pm 0.14$ |
| $\mathbb{F}_7$ | 4 | $0.09 \pm 0.14$ | $0.07 \pm 0.09$ | $0.06 \pm 0.07$ | $0.03 \pm 0.03$ | $\mathbf{0.03} \pm 0.03$ |
| | 5 | $0.18 \pm 0.41$ | $0.15 \pm 0.34$ | $0.13 \pm 0.26$ | $0.05 \pm 0.08$ | $\mathbf{0.05} \pm 0.08$ |
| | 6 | $0.28 \pm 0.54$ | $0.24 \pm 0.45$ | $0.20 \pm 0.37$ | $0.07 \pm 0.11$ | $\mathbf{0.07} \pm 0.11$ |
| $\mathbb{F}_{127}$ | 4 | $0.06 \pm 0.08$ | $0.05 \pm 0.07$ | $0.05 \pm 0.07$ | $0.02 \pm 0.02$ | $\mathbf{0.02} \pm 0.02$ |
| | 5 | $0.14 \pm 0.25$ | $0.12 \pm 0.18$ | $0.10 \pm 0.16$ | $0.04 \pm 0.05$ | $\mathbf{0.04} \pm 0.05$ |
| | 6 | $0.45 \pm 0.70$ | $0.35 \pm 0.51$ | $0.27 \pm 0.39$ | $0.10 \pm 0.11$ | $\mathbf{0.10} \pm 0.11$ |

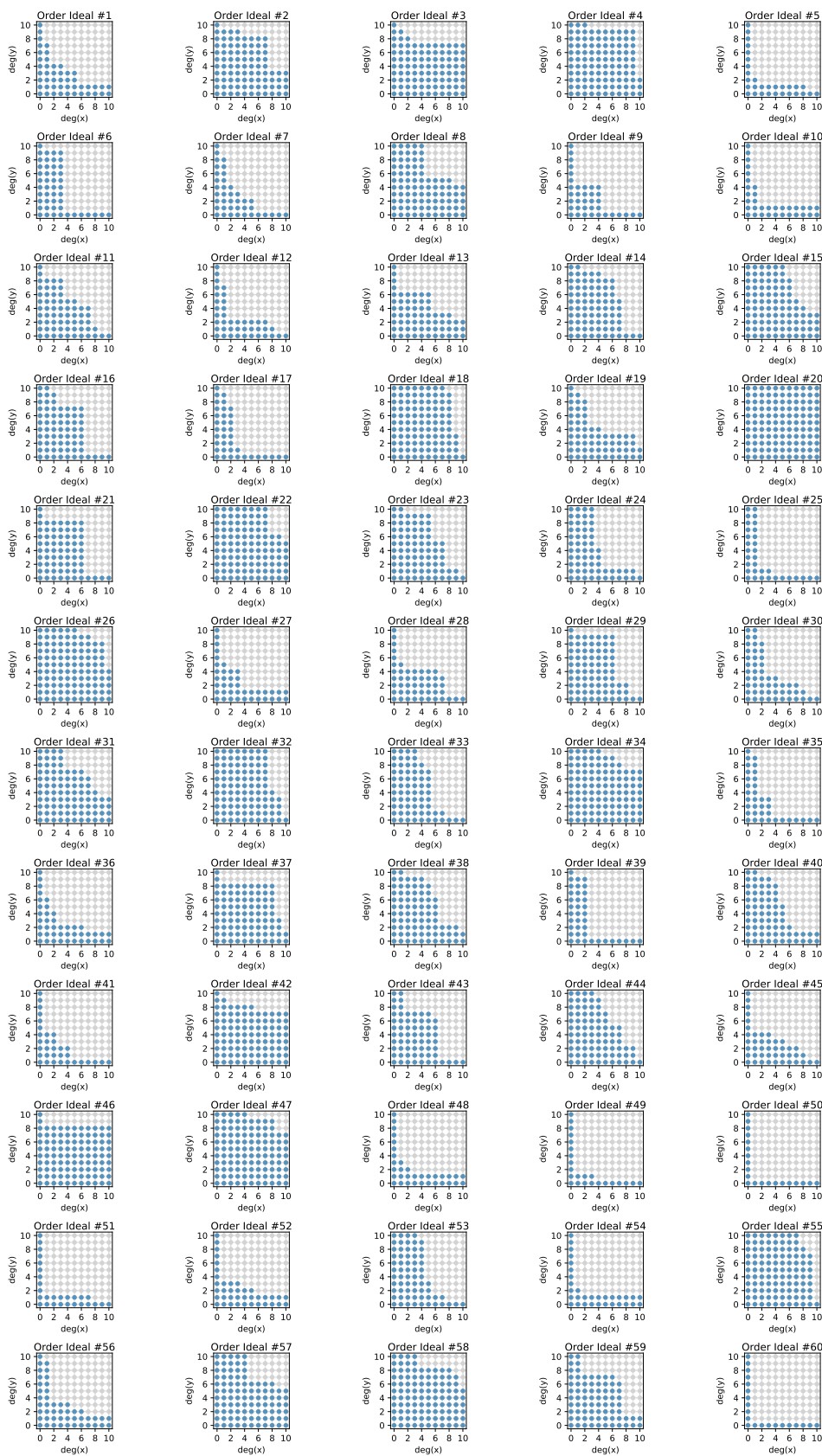

Figure 5: The gallery of randomly sampled order ideals for $n = 2$ and $\boldsymbol{d} = (10, 10)$.

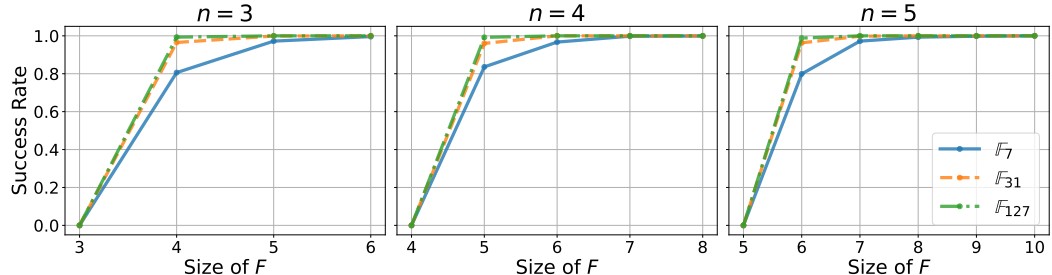

Figure 6: The empirical success rate of the backward transform from $G$ to $F$ without changing ideals. As Theorem 4.5 and Corollary 4.6 suggest, the success rate is zero for $|F| = n$ and close to one for $|F| > n$, and larger field order and number of variables increase the success rate.

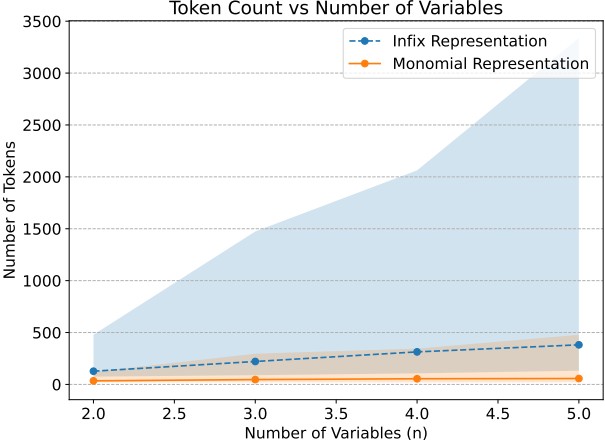

Figure 7: The average number of tokens with infix and the proposed embeddings.

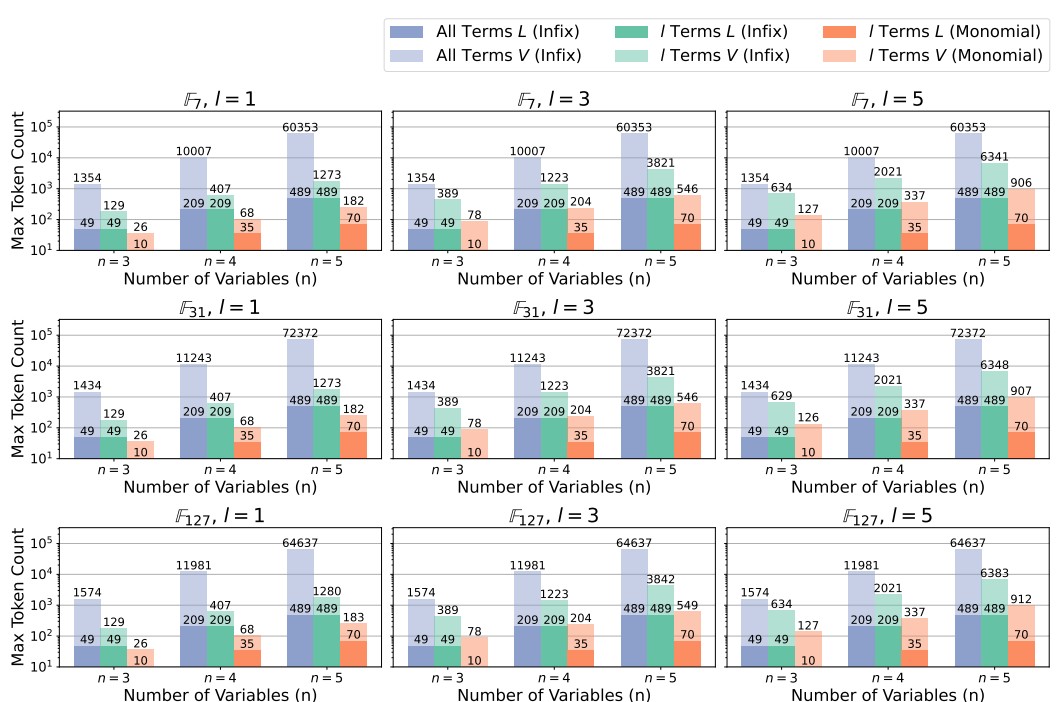

Figure 8: Reduction of the maximum number of tokens of input sequences with $l$-leading term and monomial representation.

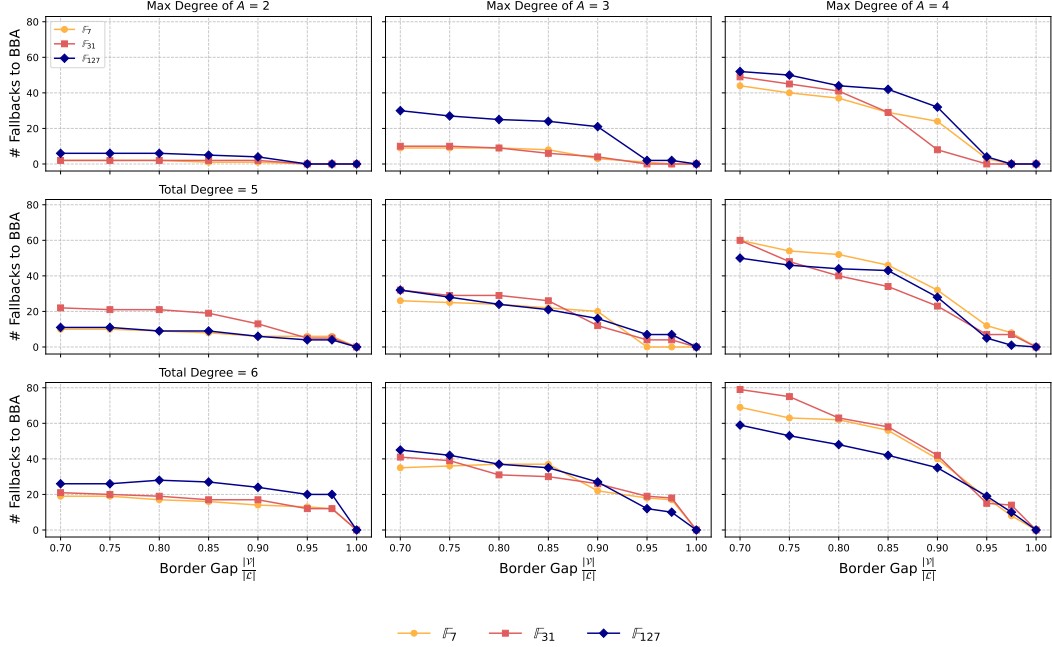

Figure 9: Out-of-distribution (OOD) experiment for $n = 3$, measured by the number of fallbacks plotted against the relative border gap. Columns correspond to the degree of the transformation matrix $A$ (left: degree 2, middle: degree 3, right: degree 4), while rows indicate the total polynomial degree, increasing from 4 in the top row to 6 in the bottom row. As the degree of the polynomials increases—moving further from the training distribution—the number of fallbacks also rises. Invoking the oracle at a lower relative border gap increases the number of fallbacks, as the oracle must make more progress to achieve termination.

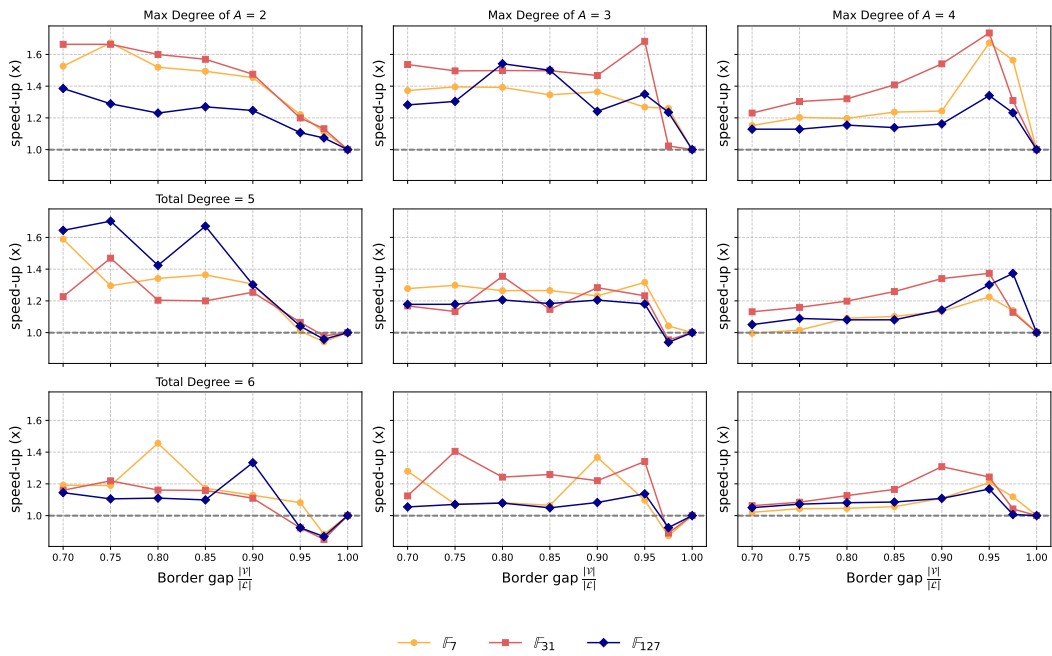

Figure 10: Out-of-distribution (OOD) experiment for $n = 3$, measured by the speedup of the proposed method over the baseline. Columns correspond to the degree of the transformation matrix $A$ (left: degree 2, middle: degree 3, right: degree 4), while rows indicate the total polynomial degree, increasing from 4 in the top row to 6 in the bottom row. Even a moderate number of fallbacks still allows for a significant speedup. For the more challenging settings, the speedup peaks around a relative border gap of $0.95$, indicating sensitivity to this parameter and reduced oracle performance in harder configurations, while still providing a moderate overall speedup.

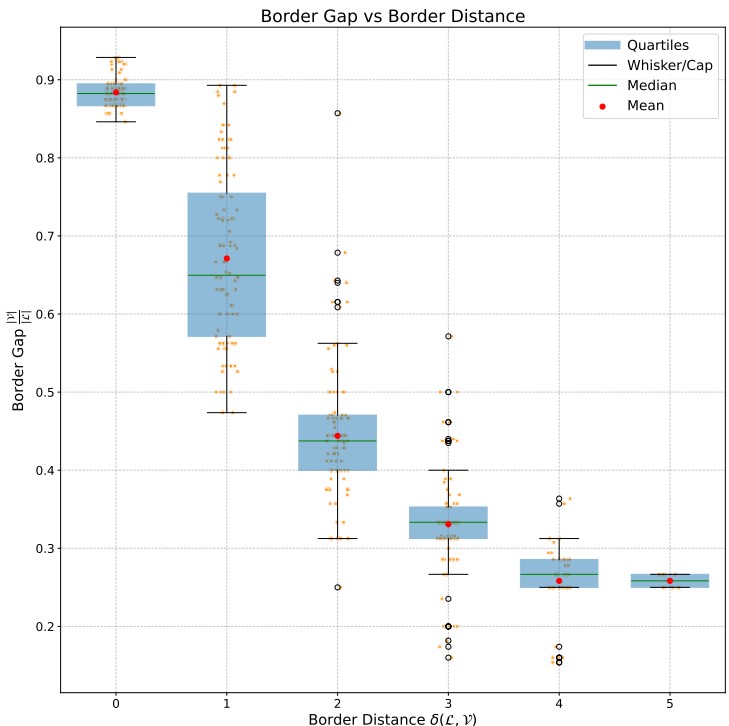

Figure 11: Border gap vs border distance for $n = 3$ over $\mathbb{F}_{31}$.

Table 10: Proportion of total runtime spent in the final stage (FS) of Border Basis computation, and cumulative share of the last $k$ expansions (L$k$) within FS, averaged over 100 runs. The final stage typically dominates runtime ($>$70%, often $>$95%), with the last 5 expansions accounting for 70–90% of FS across all settings. Interestingly, the very last expansion already account for about 30% of the final stage.

| Field | $n$ | Deg | FS | L1 | L2 | L3 | L4 | L5 |
|---|---|---|---|---|---|---|---|---|
| | | 4 | 0.986 | 0.270 | 0.516 | 0.646 | 0.699 | 0.721 |
| | 3 | 5 | 0.979 | 0.279 | 0.511 | 0.624 | 0.678 | 0.694 |
| | | 6 | 0.982 | 0.741 | 0.893 | 0.922 | 0.937 | 0.944 |
| | | 2 | 0.976 | 0.318 | 0.594 | 0.738 | 0.780 | 0.800 |
| $\mathbb{F}_7$ | 4 | 3 | 0.972 | 0.275 | 0.513 | 0.664 | 0.750 | 0.793 |
| | | 4 | 0.986 | 0.321 | 0.558 | 0.687 | 0.757 | 0.785 |
| | | 2 | 0.969 | 0.244 | 0.481 | 0.678 | 0.805 | 0.880 |
| | 5 | 3 | 0.970 | 0.270 | 0.514 | 0.704 | 0.806 | 0.862 |
| | | 4 | 0.963 | 0.301 | 0.573 | 0.768 | 0.852 | 0.889 |
| | | 4 | 0.986 | 0.285 | 0.527 | 0.652 | 0.705 | 0.721 |
| | 3 | 5 | 0.973 | 0.309 | 0.532 | 0.638 | 0.683 | 0.693 |
| | | 6 | 0.726 | 0.448 | 0.649 | 0.738 | 0.781 | 0.800 |
| | | 2 | 0.964 | 0.303 | 0.563 | 0.717 | 0.794 | 0.831 |
| $\mathbb{F}_{31}$ | 4 | 3 | 0.980 | 0.274 | 0.520 | 0.669 | 0.749 | 0.792 |
| | | 4 | 0.963 | 0.313 | 0.569 | 0.713 | 0.780 | 0.815 |
| | | 2 | 0.960 | 0.326 | 0.588 | 0.736 | 0.826 | 0.877 |
| | 5 | 3 | 0.970 | 0.275 | 0.524 | 0.714 | 0.805 | 0.858 |
| | | 4 | 0.568 | 0.289 | 0.560 | 0.744 | 0.852 | 0.895 |
| | | 4 | 0.981 | 0.287 | 0.525 | 0.655 | 0.708 | 0.727 |
| | 3 | 5 | 0.979 | 0.284 | 0.523 | 0.649 | 0.702 | 0.726 |
| | | 6 | 0.983 | 0.569 | 0.759 | 0.821 | 0.844 | 0.857 |
| | | 2 | 0.992 | 0.346 | 0.591 | 0.709 | 0.758 | 0.789 |
| $\mathbb{F}_{127}$ | 4 | 3 | 0.977 | 0.317 | 0.613 | 0.773 | 0.819 | 0.842 |
| | | 4 | 0.982 | 0.284 | 0.546 | 0.692 | 0.760 | 0.786 |
| | | 2 | 0.992 | 0.311 | 0.580 | 0.745 | 0.830 | 0.878 |
| | 5 | 3 | 0.978 | 0.279 | 0.520 | 0.692 | 0.797 | 0.866 |
| | | 4 | 0.956 | 0.290 | 0.558 | 0.746 | 0.841 | 0.891 |

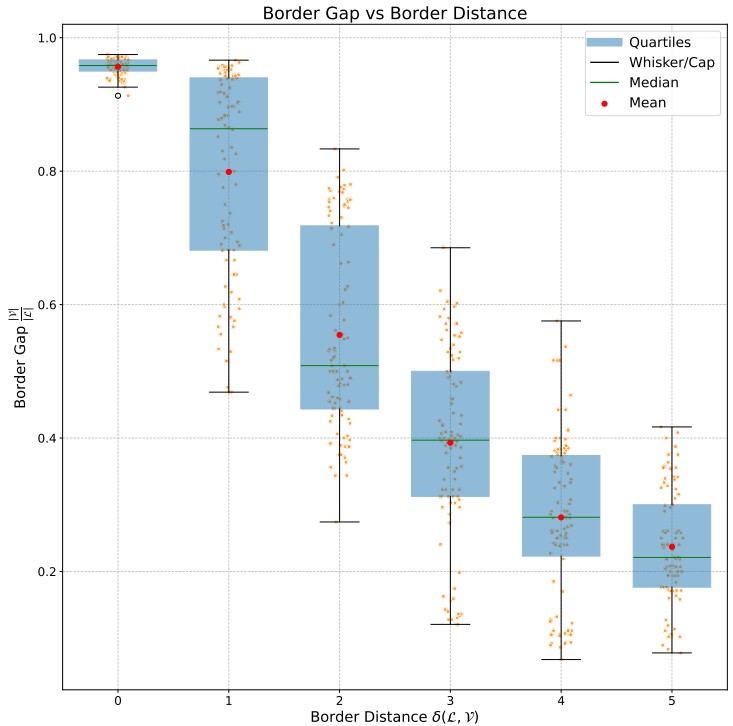

Figure 12: Border gap vs border distance for $n = 4$ over $\mathbb{F}_{31}$.

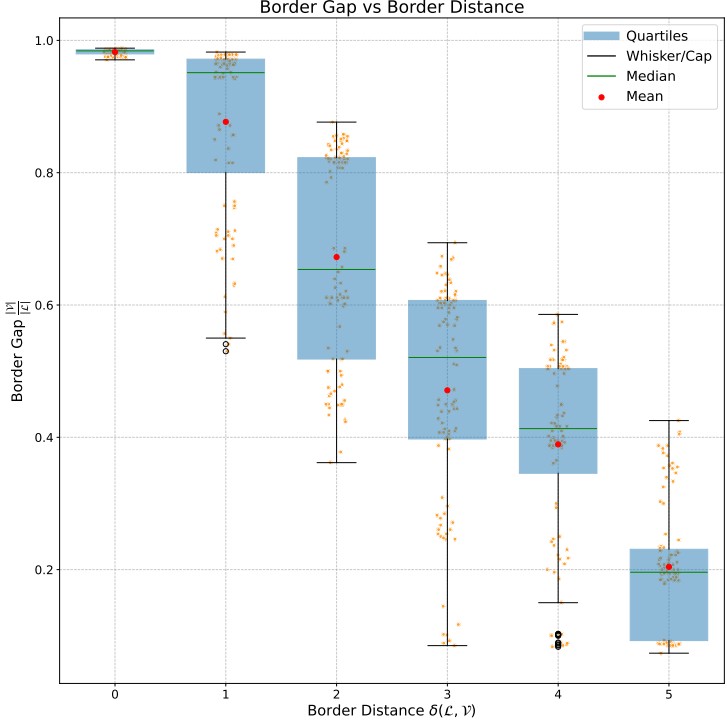

Figure 13: Border gap vs border distance for $n = 5$ over $\mathbb{F}_{31}$.

