# OpenReview forum: "Computational Algebra with Attention: Transformer Oracles for Border Basis Algorithms"
_NeurIPS.cc/2025/Conference — NeurIPS 2025 poster_

### Official Review · Reviewer_ECgM · 2025-06-30

**Clarity:** 2
**Significance:** 3
**Originality:** 2
**Rating:** 4
**Confidence:** 1

**Summary:**

This paper uses a Deep Learning approach to speedup existing
Border basis based approaches to solving systems of polynomial equations.
This can
 serve as a  practical enhancement to traditional computer algebra algorithms.
In particular, they claim to obtain  3.5x speedup compared to Border basis computer algebra algorithm which does not use AI.
Additionally, introduce a novel sampling based method to generate training data.

**Questions:**

What would be the plan going from Border bases to Grobner bases?

How does the transformer embed the polynomial input  ?

Empirical Breadth: A brief mention of the benchmarks, problem sizes, or types of systems tested would help assess practical significance.

**Ethical Concerns:**

["NO or VERY MINOR ethics concerns only"]

**Final Justification:**

The problem of solving systems of polynomial equations with finitely many solutions occurs in many fields, including cryptography, computational biology, algebraic geometry, etc. And furthermore Gröbner and border bases are part of the backbone of symbolic computation systems, although they are notoriously expensive. So, using ML to help compute Border bases is very natural.
Furthermore, using ML it in conjunction with traditional methods enables efficiency improvement without sacrificing correctness. This is a strong benefit of the approach. Finally, it seems that the speed up claimed will make the method useful in real computer algebra systems.

**Limitations:**

Not much of a discussion

**Quality:**

3

**Strengths And Weaknesses:**

Strength:

1) Natural and central problem.The problem of solving systems of polynomial equations with finitely many solutions occurs in many fields, including cryptography, computational biology, algebraic geometry, etc. And furthermore Gröbner and border bases are part of the backbone of symbolic computation systems, although they are notoriously expensive. So, using ML to help compute Border bases is very natural.

2) Using ML it in conjunction with traditional methods enables efficiency improvement without sacrificing correctness. This is a stroong benefit of the approach.

3) The speed up claimed will make the method useful in real computer algebra systems.

4 )
Tailoring tokenization and embeddings to algebraic input seems like a novel contribution in this field.


Weakness
Is 3.5 factor enough theoretically, can you do better?
how large is the experiment - I couldn't find the answer?

---

> ### Author Rebuttal · Authors · 2025-07-30
>
> We appreciate the reviewer's time and effort.
>
> While some aspects may be outside the reviewer's core expertise (judged by the consensus score), the listed strengths accurately reflect our key contributions. As these points seem to be highly appreciated by the reviewer, and as there are weaknesses that only ask for clarification, we suspect the negative score mostly comes from the uncertainty about the impact of our work in the context of the literature.
>
> Below, we highlight several strengths of this work that may be overlooked by the reviewer, and then also address the suggested weaknesses and questions. An overview is as follows.
>
> ---
> ### **Key Responses**
>
> 1. (Probably overlooked technical strengths). Our technical contributions include the **dataset generation**, which is a fundamental component in the literature of learning symbolic computation. Particularly, we highlight a) the first generic border basis sampling algorithm and b) ideal-invariant generator transform (Theorem 4.2), which significantly generalizes [Kera+'24] and offers a general method of sampling non-trivial random polynomial systems.
>
> 2. (Probably overlooked empirical strengths). The experiments demonstrate (e.g., Table 6) that our oracle leads to **large speedups (up to 30 seconds)** and **strong generalization to out-of-distribution samples**.
>
> 3. (The experiment setup: the size and type of systems). Our experiments handle **the largest and the most general class of polynomial systems in the literature** of deep learning-based Gröbner/border basis computation. This is achieved by the technical contributions highlighted in (1) and our efficient input embedding (i.e., monomial embedding).
>
> We kindly ask the reviewers to take these into account at the reassessment of this work.
>
> ---
>
> ### **(1) Technical strengths.**
> **Border basis sampling algorithm.** To our knowledge, this is the first algorithm that allows us to sample generic border bases. Recall that border bases are a generalization of Gröbner bases in the zero-dimensional case. Only border bases in a special (measure-zero) subset have associated Gröbner bases. An existing method [Abbott+'05] only offers sampling from such a special subset, but our algorithm can sample general border bases.
>
> **Ideal-invariant generator transform (Theorem 4.2)** This offers a principled design of a polynomial matrix A that transforms the polynomial set G by F=AG without changing the ideal (i.e., \<F\> = \<G\>). [Kera+'24] only covers the extension (i.e., |F| >= |G|), but ours addtional cover the contraction (|F| < |G|). The design of A is also significantly simplified.
>
> ---
> ### **(2) Empirical strengths.**
> Our experiments, both in the main text and the Appendix, not only focus on the speedups but also examine the generalizability of the oracle to OOD instances. In case the reviewer judges the empirical results only from the main text, we highlight the results in Table 6 from the Appendix. As shown in Table 6 (bottom row, IBB vs. IBB+Oracle), our oracle achieves speedups of up to 30 seconds on hard, out-of-distribution instances. These results also demonstrate strong generalization to OOD inputs.
>
>
>
> > Is 3.5 factor enough theoretically, can you do better?
>
> We appreciate the reviewer's question. As shown in Table 5 (Appendix), the observed 3.5× speedup is already close to the best achievable. A perfect oracle—one that never predicts a polynomial reducible to zero—would reduce the number of unnecessary polynomials from the IBB baseline of roughly 1300–1400 down to none. Our oracle typically predicts only 300–400 such polynomials, which matches the 3.5× improvement. Even a perfect oracle could only achieve around 5× speedup, so our result is already close to this theoretical limit.
>
>
> ---
> ### **(3) The experiment setup.**
> > Empirical Breadth: A brief mention of the benchmarks, problem sizes, or types of systems tested would help assess practical significance.
>
> To the best of our knowledge, our experiments cover the largest and the most general class of polynomial systems in the literature of deep-learning based Gröbner/border basis computation [Peifar+'20, Kera+'24].
> - The experiments in [Peifar+'20] mostly focus on n=3 variables and binomials (for both input systems F and Gröbner bases G).
> - The experiments in [Kera+'24] handle up to n=5, but for n>1, the mixing matrix A of F=AG is sparsified to keep the input system size moderate (i.e., to keep the number of tokens < 5000). The design of A only supports an input system of size $|F| >= |G|$. Besides, the ideals are restricted to those in shape position, where G is restricted to size |G|=n.
>
> The following table compares the scale and class of systems and ideals (n=5, numbers are rounded).
> |                | Input system size \|F\| | Avg # of terms in F | class of ideals and systems | Note                                                  |
> |----------------|-------------------------|----------------------|----------------------| -------------------------------------------------------|
> | [Peifer+’20]   | 10                      | 20                   | binomial |  Most experiments were done with n=3.
> | [Kera+'24]     | 5–7                     | 42                   | shape position (\|F\| >= \|G\| = n) | From "Max terms in F" in Table 6 in [Kera+'24].       |
> | Ours           | 6–10                    | 130                  | vanishing ideal (\|G\| >= n, OK with \|F\| < \|G\|) | Computed from our datasets, will be added in Appendix.|
>
> We acknowledge that we currently focus on small systems, which may not have immediate benefits in real-world applications. However, this work should achieve large progress in the field of deep learning-based Gröbner/border basis computation through new algebraic and machine learning techniques (e.g., oracle-based technique with certified output, ideal-invariant transformation, monomial embedding), which will generalize to deep learning-based computer algebra.
>
> ---
> ### **(4) Others**
> **Q1. From Border Bases to Gröbner bases.**
> > What would be the plan going from Border bases to Gröbner bases?
>
> The adaptation of our oracle-based approach to compute Gröbner bases remains an interesting open problem. While this is beyond the scope of this work, and border basis computation already addresses the polynomial system solving task in the zero-dimensional case, we share our view on this adaptation.
>
> One key observation in our work is that the border basis algorithm proceeds degree-by-degree, and one can determine successful/unsuccessful expansions in hindsight. This enables us to train the Transformer-oracle in a supervised manner. Thus, it is critical for the adaptation to find such computational steps in computing Gröbner bases. In preliminary analysis, such hindsight-verifiable structures do not appear in standard Gröbner basis algorithms (e.g., Buchberger's, F4), where computation cost depends heavily on reduction ordering. For future plans, one can examine less popular algorithms of Gröbner basis computation, or design a new one so that it is more straightforward to define an oracle.
>
>
> **Q2. Embedding polynomials.**
> > How does the transformer embed the polynomial input?
>
> We here give an intuition using a polynomial $f = 2x^3 + y + 1 \in \mathbb{F}_7[x, y]$ as a running example.
>
> A polynomial is represented in a sum of monomials, and first tokenized as (C2, E3, E0, +, C1, E0, E1, +, C0, E0, E0, \<\*\>), where Cn and Em denotes coefficient $n$ and exponent m, respectively, and \<\*\> is a token that follows this polynomial (e.g., \<sep\>, \<supsep\>, \<eos\>).
>
> The standard embedding method simply translates each of the 12 tokens above into the associated embedding vector in the trainable look-up matrix (implemented by `nn.Embedding` in PyTorch). Thus, the embedding of $f$ becomes a sequence of 12 embedding vectors. In essence, the proposed embedding  (i.e., monoial embedding method) sums up the embedding vectors inside each monomial and its follow-up token. Namely, (C2, E3, E0, +, C1, E0, E1, +, C0, E0, E0, \<\*\>) is reduced to 3 embedding vectors, where for example, the first embedding vector is the sum of embedding vectors of (C2, E3, E0, +). Note that we use different look-up matrices for exponents of x and y so that we can distinguish (C2, E3, E0, +) and (C2, E0, E3, +) after adding up embedding vectors. Section 4.2 and Appendix D provide the full description of the proposed embedding method.
>
> Table 3 demonstrates that the monomial embedding not only reduces the number of tokens (i.e., memory cost in attention) but also improves the success rate. Table 7 shows how significantly our training benefits from this embedding. As suggested by Reviewers CaQ9 and C42S, monomial embedding is one of the key technical contributions of this work, and is useful for learning other polynomial-based tasks.
>
> ---
> - [Peifer+'20] Peifer et al. "Learning Selection Strategies in Buchberger's Algorithm," ICML'20
> - [Kera+'24] Kera et al., "Learning to Compute Gröbner bases," NeurIPS'24

---

> > ### Comment · Reviewer_ECgM · 2025-08-04
> > **Comment post Rebuttal**
> >
> > Thank you for the clarification. The experimental results are better than I understood originally.
> > I will update my score to border line accept, with low confidence....once I understand how to do it on this website.

---

> > > ### Author Response · Authors · 2025-08-05
> > > **Acknowledgment for reassessment**
> > >
> > > We are grateful for the reviewer’s thoughtful reassessment and plan to update the score. As noted, some aspects of the experimental results may have been initially underappreciated. To avoid any potential misunderstandings, we will make sure to clarify these points thoroughly in our next revision.

---

### Official Review · Reviewer_C42S · 2025-07-03

**Clarity:** 3
**Significance:** 3
**Originality:** 3
**Rating:** 4
**Confidence:** 4

**Summary:**

This paper introduces the Oracle Border Basis Algorithm (OBBA), the first deep learning approach that accelerates border basis computation for solving polynomial equation systems while maintaining mathematical correctness guarantees.

**Questions:**

See Weakness and Limitation sessions

**Ethical Concerns:**

["NO or VERY MINOR ethics concerns only"]

**Final Justification:**

While I find the author’s reply satisfactory, I will retain my initial score and decision, as the overall quality does not compare favorably with other papers I reviewed.

**Limitations:**

This paper introducing the first deep learning approach that successfully accelerates border basis computation while maintaining mathematical correctness guarantees. The Oracle Border Basis Algorithm represents a pioneering achievement in AI-enhanced symbolic computation, demonstrating up to 3.5× speedup with rigorous mathematical proofs and novel monomial embedding techniques.
However, several critical practical challenges severely limit the real-world applicability of the proposed solution. The method requires generating millions of training samples for each new problem domain, creating substantial computational overhead that may offset the runtime benefits in practice. The evaluation lacks comparison with other acceleration techniques for symbolic computation, making it difficult to assess the relative merit of the ML-based approach versus existing optimization methods. The training is limited to degree 2 systems while testing shows only modest generalization to degree 8, raising questions about scalability to higher-degree polynomial systems commonly encountered in real applications. The restriction to finite fields, "leaving characteristic-0 arithmetic and larger arity unexplored," significantly constrains the method's applicability to many practical domains that require real or complex number computations. Additionally, the experimental evaluation is based on only 27 constructed datasets with limited parameter variations, providing insufficient evidence for the method's robustness across diverse problem structures and raising concerns about whether the observed performance gains will generalize to the broader landscape of polynomial system solving tasks.

**Quality:**

3

**Strengths And Weaknesses:**

Strengths:
Successfully combines deep learning with symbolic computation while maintaining mathematical correctness.
Monomial embedding reduces token count by O(n) and attention memory by O(n²).
Clever insight that border basis computation's degree-by-degree structure enables more stable supervised learning vs. reinforcement learning.

Weakness:
Requires generating millions of training samples for each new problem domain.
No comparison with other acceleration techniques for symbolic computation.
Training only on degree 2 systems, though testing shows some generalization to degree 8
Finite fields only: "leaving characteristic-0 arithmetic and larger arity unexplored"
Only 27 datasets constructed.

---

> ### Author Rebuttal · Authors · 2025-07-30
>
> We thank the reviewer for the thoughtful assessment. We recognize the reviewer's main concern as the practical limitations that may affect the applicability of our method. We partially acknowledge these limitations but also would like the reviewer to comprehend the development of the field for assessment. Particularly, we first highlight the following.
>
> ---
> ### **Key Responses**
>
> - Solving polynomial systems has been a long-standing, hard task. Our experiments handle **the largest and most general class of polynomial systems in the literature** of deep learning-based Gröbner/border basis computation. We believe that this work should make large progress from prior studies toward real-world applications, **through new algebraic and machine learning techniques** (e.g., oracle-based technique with certified output, ideal-invariant transformation, monomial embedding).
>
> - Finite field systems cover broader cases than the reviewer may consider. Indeed, from an algebraic perspective, **it is common to reduce infinite-field systems (including real and complex numbers) to finite-field systems** using the modular technique. Further, from a machine learning perspective, **learning on an infinite field is well-known significantly easier**, which motivated us to focus on the finite field.
>
> - Besides 27 datasets for training and in-distribution evaluation, we have **27 out-of-distribution evaluation sets** for examining the generalizability of the trained oracles, which should possess sufficient variations to support our claim.
>
> ---
> **W1. Instance sampling for new problem domains.**
> > The method requires generating millions of training samples for each new problem domain, creating substantial computational overhead that may offset the runtime benefits in practice.
>
> We do not consider this to be a particular weakness of our study. This criticism can be applied to various deep learning studies. While generalization is important, for new domains, it is common to collect new samples and retrain (or fine-tune) models on them. In the context of machine learning oracle-based algorithms, it is common to collect many training instances by running the algorithms.
>
> We emphasize two overlooked aspects: (1) training data collection is less costly than it may seem, and (2) the oracle achieved significant runtime reductions on harder, out-of-distribution instances.
>
> 1. Collecting one million samples does not mean running algorithms one million times. Border basis algorithms involve many expansion steps, each of which can be collected as training instances. In our case, we focus on the final k=5 calls of expansions,  which are very costly and thus benefit from the oracle. So, we need 200k runs. The value of T is selected from our analysis of algorithm runs on our problem size. This k will be set larger for harder instances because there will be more "costly expansions" in a single run.
>
> 2. The trained oracle empirically generalized well to harder and out-of-distribution instances. Particularly, Table 6 (see the bottom row, IBB vs IBB+Oracle) presents that Oracle reduces the runtime by approximately 30 seconds on average for hard instances that are also out of distribution. Note that FGE is also our contribution, which is a byproduct of this work, and it should be fair to compare the impact of the oracle without FGE.
>
> ---
>
> **W2. Other acceleration techniques for symbolic computation**
> > The evaluation lacks comparison with other acceleration techniques for symbolic computation, making it difficult to assess the relative merit of the ML-based approach versus existing optimization methods.
>
> We compared our method to the standard Border Basis Algorithm (BBA) as well as the SOTA Improved Border Basis Algorithm (IBB) and introduced our own acceleration technique in the form of fast Gaussian elimination. We may be able to collect more techniques from other symbolic computation algorithms to accelerate border basis algorithms, but this should form an independent work of fast border basis computation. Therefore, we consider that our choice of baselines should be reasonable.
>
> ---
> **W3. Training only on degree 2 systems.**
> >  The training is limited to degree 2 systems while testing shows only modest generalization to degree 8, raising questions about scalability to higher-degree polynomial systems commonly encountered in real applications.
>
> This involves a misunderstanding. The input systems F in training instances range up to degree 3, as these are obtained F=AG with border bases G of degree-2 and random polynomial matrices A, which produce a wide range of degrees in F. We kept the degree low to see how the oracle generalizes to higher degrees. In Figs. 3 and 8 and Tables 6-8 show the great generalization to various maximum degrees the input systems ranging from degree 2 to 8.
>
> The following table presents that our work addresses the largest and most general class of systems among the related studies.
> |                | Input system size \|F\| | Avg # of terms in F | class of ideals and systems | Note                                                  |
> |----------------|-------------------------|----------------------|----------------------| -------------------------------------------------------|
> | [Peifer+’20]   | 10                      | 20                   | binomial |  Most experiments were done with n=3.
> | [Kera+'24]     | 5–7                     | 42                   | shape position (\|F\| >= \|G\| = n) | From "Max terms in F" in Table 6 in [Kera+'24].       |
> | Ours           | 6–10                    | 130                  | vanishing ideal (\|G\| >= n, OK with \|F\| < \|G\|) | Computed from our datasets, will be added in Appendix.|
>
>
> ---
> **W4. Finite fields only.**
> > The restriction to finite fields, "leaving characteristic-0 arithmetic and larger arity unexplored," significantly constrains the method's applicability to many practical domains that require real or complex number computations.
>
> While the reviewer suggests the restriction to finite fields is a significant constraint, we would like to emphasize that this is not, from an algebraic and machine learning perspective. We will update the line "leaving characteristic-0 arithmetic and larger arity unexplored" to avoid confusion.
>
> 1. (Algebraic perspective). It is common to reduce the infinite-field case to the finite-field case. The technique is called *modular techniques* [Arnold'03]. While the standard modular technique assumes homogeneous systems, the homogenization trick turns any system into a homogeneous one. For example, a complex polynomial system is reduced to a real polynomial system by introducing $i = \sqrt{-1}$ as a new variable and appending $i^2 + 1 = 0$ to the system. The new system is then homogenized and processed by the modular technique (note: we also has to replace $\mathbb{R}$ to $\mathbb{Q}$ as $\mathbb{R}$ is not an algebraic field).
>
> 3. (ML perspective). Learning on a finite field has been known to be significantly more challenging than on an infinite field. For example, the parity problem (input: bit strings, target: sum of bits modulo 2) has been known to be notoriously hard to learn because of the "modulo 2" operation [Shwartz+'17, Hahn+'24]. In a more related context, [Kera+'24] reports that learning to compute Gröbner bases is less successful with finite fields (success rate 93.7% for Q vs 46.8% for F31, Table 2 in [Kera+'24]). Thus, our results should readily extend to infinite fields. We will update the manuscript to clarify that we focus on finite fields, as it has been known as a challenging case.
>
> ---
> **W5. Only 27 datasets.**
> > Additionally, the experimental evaluation is based on only 27 constructed datasets with limited parameter variations, providing insufficient evidence for the method's robustness across diverse problem structures and raising concerns about whether the observed performance gains will generalize to the broader landscape of polynomial system solving tasks.
>
> We believe that our 27 datasets are enough in numbers and diversity to validate the benefits from oracle, e.g., the speedups and the generalization to out-of-distribution samples. The number 27 only accounts for the training sets, and we additionally used 27 out-of-distribution sets in evaluation as shown inTables 6-8], which is from the combinations of 3 fields, 3 variable sizes (n=3,4,5), 3 max degrees in G.  (note: to be precise, these 27 have a small overlap with the original, but instead we have the same number of additional patterns used in Figs. 8. So, the total number of out-of-distribution sets is 27.)
>
> Each dataset was regenerated with many random seeds, coefficient scales, and sparsity patterns—so a single family already spans runtimes from under 10 s to over 1000 s (cf. Table 6). Despite being trained only on degree-up-to-3 systems, the oracle generalizes to degree 4 and 8 (Fig. 3) and to the 12 additional families reported in the appendix (Figures 8-9), all with the same performance trends (Table 6-8). These benchmarks therefore provide strong evidence that the method is robust across a broad spectrum of polynomial-system problems.
>
>
> ---
> - [Peifer+'20] Peifer et al., "Learning Selection Strategies in Buchberger's Algorithm," ICML'20
> - [Kera+'24] Kera et al., "Learning to Compute Gröbner bases," NeurIPS'24
> - [Arnold'03]  Arnold, "Modular algorithms for computing Gröbner bases," JSC, 2003
> - [Shwartz+'17] Shwartz et al., "Failures of Gradient-Based Deep Learning," ICML'17
> - [Hahn+'24] Hahn et al., "Why are Sensitive Functions Hard for Transformers?," ACL'24

---

### Official Review · Reviewer_CaQ9 · 2025-07-03

**Clarity:** 3
**Significance:** 3
**Originality:** 3
**Rating:** 4
**Confidence:** 3

**Summary:**

Solving systems of polynomial equations is a foundational problem in mathematics and many other domains. Classical methods like Gröbner and Border bases provide a way to search for solutions but are computationally intensive. The paper introduce the Oracle Border Basis Algorithm (OBBA), which provides a way to efficiently searching solutions for zero-dimensional cases. They use a lightweight Transformer oracle to predict and eliminates unhelpful polynomial reductions, which significantly reduce the computational overhead.

**Questions:**

please see the weakness section.

**Ethical Concerns:**

["NO or VERY MINOR ethics concerns only"]

**Final Justification:**

Thank you for your response. I keep my score.

**Limitations:**

yes.

**Paper Formatting Concerns:**

no.

**Quality:**

3

**Strengths And Weaknesses:**

Pros
1. The idea is novel s.t combining ML/DL with rigorous math problem is relatively new and rare in the domain.
2. They designed an interesting way to train a Transformer with customized tokenization and embedding.
3. The method shows a noticeable increasing in speed without compromising accuracy and can be generalized to OOD.

Cons
1. The method is only constrained to zero-dimensional problems and can’t be generalized to positive-dimensional cases.
2. The decision of when to invoke the oracle is heuristic and could benefit from theoretical analysis or learned policies.
3. Even though the experiment shows results up to 5 variables, but the scalability to higher-dimensional or real-world symbolic problems remains unclear. Has no theoretical guarantee.
4. It would be interesting to know if we randomly prune the search tree, how would the performance looks like.

---

> ### Author Rebuttal · Authors · 2025-07-30
>
> We are grateful for the reviewer's thoughtful comments and the clear articulation of both strengths and limitations. Below, for clarity, we number the weaknesses (W1–W4).
>
>
> ---
> **W1. Constrained to zero-dimensional problems.**
> > The method is only constrained to zero-dimensional problems and can't be generalized to positive-dimensional cases.
>
> This study focuses on zero-dimensional ideals, and the positive-dimensional case is beyond its scope (border bases are only defined for the zero-dimensional case). We do not view this as a critical limitation, as **zero-dimensional systems are an important and broadly applicable subclass**, perhaps more than it first appears (see below). It is also worth noting that the contribution of this study includes monomial embedding, a general embedding method to feed large-scale systems to the Transformer.
>
> 1. All finite-field systems (let K be the field) can be reduced to the zero-dimensional case by including *field equations* (e.g., x(x-1) and y(y-1) when K = F_2[x, y]), which restrict the solution spaces to K.
>
> 2. Solving infinite-field systems is often addressed by *modular techniques* [Arnold'03] that reduce the systems to finite-field ones. While the standard modular technique assumes homogeneous systems, the homogenization trick turns any system into a homogeneous one.
>
> Therefore, **in essence, polynomial system solving can be reduced to solving zero-dimensional systems in finite fields**, although this reduction may not always be the most efficient strategy.
>
> To the best of our knowledge, [Peifer+'20] is the only work that addressed positive-dimensional ideals in the literature of deep learning-based Gröbner/border basis computation, but they instead assume the input and output systems consist of binomials, which we believe is stronger than the zero-dimensional assumption. The following table compares the scale and class of systems and ideals (n=5, numbers are rounded) in the related studies.
> |                | Input system size \|F\| | Avg # of terms in F | class of ideals and systems | Note                                                  |
> |----------------|-------------------------|----------------------|----------------------| -------------------------------------------------------|
> | [Peifer+’20]   | 10                      | 20                   | binomial |  Most experiments were done with n=3.
> | [Kera+'24]     | 5–7                     | 42                   | shape position (\|F\| >= \|G\| = n) | From "Max terms in F" in Table 6 in [Kera+'24].       |
> | Ours           | 6–10                    | 130                  | vanishing ideal (\|G\| >= n, OK with \|F\| < \|G\|) | Computed from our datasets, will be added in Appendix.|
>
>
> The extension to the positive-dimensional case is an important future work. However, this requires an efficient sampling of (non-Gröbner basis, Gröbner basis) pairs, which is an open problem.
>
> ---
> **W2. Oracle invocation.**
> >The decision of when to invoke the oracle is heuristic and could benefit from theoretical analysis or learned policies.
>
> We found that the relative border gap, defined as $\frac{|\mathcal{V}|}{|\mathcal{L}|}$, is a practical indicator for when to invoke the oracle (see Figure 3). The intuition comes from Lemma A.2: when $|\mathcal{V}|$ is close to $|\mathcal{L}|$, only a few expansions remain. To our knowledge, there are no theoretical results that prescribe the optimal point for oracle invocation during Border basis computation. However, we tested this heuristic extensively against the actual number of remaining expansions (Figures 10–12) and consistently found a strong correlation across all setups.
>
> While learned policies could further improve the decision rule, we observed that a relative border gap of about $0.9$ is nearly always optimal or very close to optimal (Figures 3, 8–9). Lemma A.2 also provides a rigorous proof for the special case where no expansions are left, but the IBB still performs one costly, full expansion. In these cases, the Transformer is especially good at predicting this final expensive step (Table 4).
>
> ---
> **W3. Scalability to higher dimensions and real-world cases**
> > Even though the experiment shows results up to 5 variables, but the scalability to higher-dimensional or real-world symbolic problems remains unclear. Has no theoretical guarantee.
>
> We do scale our case of 5 variables up to degree 4, for which the SOTA requires on average 100 seconds, whereas our pipeline (Oracle + Fast Gaussian Elimination) results in speedups of roughly **17x**. We note that our improvements are orthogonal to the baseline IBB - in particular, a highly optimised version of IBB would likely still enjoy a speedup of similar magnitudes.
>
> ---
> **W4. Random pruning.**
> > It would be interesting to know if we randomly prune the search tree, how would the performance looks like.
>
> We appreciate an interesting suggestion. We examined this idea and concluded that a purely random cut will almost certainly discard at least one mandatory reduction step, it delivers essentially no speedup.
>
> In our experiments (Table 5) the search space contains 1,000–3,000 candidates, yet only 1–15 % are needed to build a border basis. Missing even one of them derails the computation. For example, let an elimination stage produce N = 1000 candidates, with about m = 20 being mandatory (as is typically seen in the last expansion). Keeping exactly half (k = 500) succeeds only if all m survive, and the success rate P can be computed as
>
> $P=\frac{\binom{N-m}{k-m}}{\binom{N}{k}}=\frac{\binom{980}{480}}{\binom{1000}{500}}\approx7.9\times10^{-7}$.
>
> ---
>
> - [Arnold'03]  Arnold, "Modular algorithms for computing Gröbner bases," J. Symb. Comput. 35, 403-419 (2003)
> - [Peifer+'20] Peifer et al. "Learning Selection Strategies in Buchberger's Algorithm," ICML'20
> - [Kera+'24] Kera et al., "Learning to Compute Gröbner bases," NeurIPS'24

---

> > ### Author Response · Authors · 2025-08-06
> > **Discussion Closing**
> >
> > Dear Reviewer,
> >
> > We kindly remind you that the discussion period is coming to a close. We believe that our response addresses your concerns, particularly by clarifying several potential misconceptions and providing clearer contrasts with related work.
> >
> > We understand that our rebuttal is quite detailed, so we would sincerely appreciate it if you could take a moment to review at least the **Key Responses** section.
> >
> > Thank you very much for your time and consideration.

---

### Official Review · Reviewer_y4oj · 2025-07-11

**Clarity:** 3
**Significance:** 3
**Originality:** 3
**Rating:** 4
**Confidence:** 3

**Summary:**

The paper presents OBBA, an algorithm for solving zero‐dimensional systems, which integrates a Transformer oracle into border‐basis computations. The authors show that OBBA has reduced wall-clock runtime, while preserving nearly perfect accuracy. The speed-up is obtained by using the oracle to predict only the necessary expansions in the final steps of the border-basis algorithm.

**Questions:**

- Can you comment on the originality of the proof techniques used?
- Can you explain why you chose the number of variables and degree to be up to 5 and 4, respectively, and how you would expect the runtime gain to scale?
- Can you comment on how you chose the Transformer architecture and on whether you did any ablation about that?
- In Figure 3, what is the OOD accuracy achieved? Can you also comment on the trend of the number of fallbacks for degree 4 versus degree 8?

I believe there are small typos in line 103 (T_n missing) and in line 94 (caligraphic I?).

**Ethical Concerns:**

["NO or VERY MINOR ethics concerns only"]

**Final Justification:**

The authors clarified the technical novelty and experimental results in their work during rebuttal.

**Limitations:**

The algorithm proposed is limited to finite-sized polynomial systems, since it relies specifically on the border-basis algorithm.

**Quality:**

3

**Strengths And Weaknesses:**

Strengths:
- The paper is well written, and contains a complete overview of the techniques used and comparison to state of the art.
- Proposing new faster algorithms for symbolic computation is an important direction. The idea of integrating a Transformer-based oracle and analysing its complexity is nice and promising.

Weaknesses:
- My main concern is that the scope of this work might be a bit narrow.
- To the best of my understanding, the theoretical proofs, while sound, they assemble previously known facts. I would be happy to change my assessment if the authors do not agree and can elaborate more on the originality of their proof techniques.
- It seems therefore that the main contribution consists of the reduced wall-clock runtime presented in the experiments, along which the reduction of the input size for making the problem tractable by a Transformer. However, the experiments are for fairly small number of variables and degree. The empirical speedups, although measurable, occur on very small instances (wall-clock times on the order of 10⁻² s). It would be helpful to understand the limitations or bottlenecks when scaling to larger systems
- It remains unclear which real-world applications benefit from these runtime improvements on such small systems; I invite the authors to elaborate more on these.

---

> ### Author Rebuttal · Authors · 2025-07-30
>
> We appreciate the reviewer's insightful and thorough review. We understand the main concerns are: (1) Lack of technical novelty in the proofs, (2) Limited runtime gain with our oracle-based approach, and (3) Unclear benefit in the real-world applications.
>
> We respectfully disagree with the first two criticisms and ask the reviewer to consider the broader context of research development when evaluating the third. The short answers are as follows.
>
> ---
>
> ### **Key Responses**
>
> 1. The theoretical results do consist of the main contributions of our work - the claims are all novel, and the proofs are **non-trivial assembly of known facts** (Any "novel proofs in math" assemble known facts.).
>
> 2. The best empirical average runtime gain by the oracle is **approximately 30 seconds, not $10^{-2}$ as the reviewer claims**.
>
> 3. We acknowledge that success in small systems may not have immediate benefits in real-world applications. Nevertheless, our oracle-based approach and novel algebraic and machine learning techniques tackle **the largest and most general systems in the literature** - a critical step toward real-world deployment.
>
> We kindly ask the reviewer to reassess our contributions (particularly technical ones) and their position in the literature.
>
> ---
> ### (1) Technical novelty in the proofs
> > [...], the theoretical proofs, while sound, they assemble previously known facts.
>
> > Can you comment on the originality of the proof techniques used?
>
> It is unclear to us which of the proofs the reviewer finds lacking technical novelty. Our work contains three main theoretical contributions. Each of them is novel in problem formulation, and the proofs go beyond a simple assembly of known facts: a) **Analysis of the Oracle Border Basis Algorithm** (Thm 3.1 and Sec. A). b) **Border Basis Sampling** (Thm. 4.2 and App. B). c) **Ideal-Invariant Generator Transform** (Thm. 4.5 and App. C).
>
> The reviewer probably refers to (c), as it prepares Lemma C5-8 with citations. However, these except for C8 are all textbook-level lemmas, included to ensure clarity and accessibility for readers who are less familiar with abstract algebra. Our proof technique itself is original in that it non-trivially combines the theory of parameterized ideals with primary decomposition.
>
> For completeness, we address (a) and (b).
>
> a) Our novel concepts (e.g., the *border distance*; Def. A.3) enable the definition of a *conditional error* (Def. A.4), which is pivotal in the proof to connect the prediction error and the cost relative to the optimal expansion sequence. The result is not only theoretically significant but also serves as the primary motivation for invoking the oracle only at the final stages of the border basis computations.
>
> b) Border basis sampling is a novel task, and we propose an algorithm to address this. The proof of Theorem 4.2 is its basis. While the proof is not very complex, it solidly derives the arguments by exploiting the definition and characteristics of border bases.
>
> Results (a), (b), and (c) form key contributions of this work. Particularly, (c) extends beyond border basis computation, offering a principled method for generating non-trivial random polynomial systems.
>
> We hope the reviewer will take into account the above points in reaccessment.
>
> ---
> ### (2) The empirical speedup and OOD accuracy.
> > [...] the main contribution consists of the reduced wall-clock runtime presented in the experiments, [...]. The empirical speedups, although measurable, occur on very small instances (wall-clock times on the order of 10⁻² s).
>
> The reduced wall-clock runtime is only a part of our contributions; we have many technical contributions, partially elaborated above.
>
> Besides, it seems that the reviewer is underrating the runtime gain by the oracle. As shown in Table 6, the best average runtime gain goes up to approximately 30 seconds (IBB vs IBB+Oracle in the bottom row). Note that FGE is also our contribution and a byproduct of this work; thus, it is fair to compare the impact of the oracle independently of FGE.
>
> Further, our empirical contributions include the generalization of the oracle to the hard, out-of-distribution instances. See below.
>
> > [...] why you chose the number of variables and degree to be up to 5 and 4, respectively, and how you would expect the runtime gain to scale?
>
> > In Figure 3, what is the OOD accuracy achieved? Can you also comment on the trend of the number of fallbacks for degree 4 versus degree 8?
>
> > It would be helpful to understand the limitations or bottlenecks when scaling to larger systems.
>
> We appreciate these in-depth questions.
>
> We follow the Gröbner evaluation grid of 5 variables × degree ≤ 4 in [Kera+'24].  Tab. 2 already shows that the classical IBB solver's runtime increases dramatically from 0.06 s (n = 3, d = 4) to 7.6 s (n = 5, d = 2)—an ≈100× increase for just two extra variables. To probe harder cases, we keep n = 5 but raise the degree to 4; this OOD suite pushes IBB to 97.8 s mean / +1000 s max per instance (cf. Table 6).  **OBBA still removes 30 % of those reductions, and—when paired with our fast Gaussian-elimination, the wall-clock time to 5.63 s (17× speedup / 90s speedup)**. The scaling is consistent across $n = 3, 4, 5$ and $d = 2, 3, 4, 5, 6$; see Table 6-8. We expect the runtime gain to scale similarly to larger instances, as the dataset generation would become more expensive.
>
> The OOD accuracy in Fig. 3 depends on how aggressively we use the oracle. For degree 4, the success rate reaches 95%. For degree 8, which is further from the training data, accuracy starts around 60% but quickly rises above 95% with more conservative oracle use. For degree 4, calling the oracle earlier gives larger speedups because the oracle is highly reliable. For degree 8, the best speedup happens when the oracle is invoked near the border gap $\frac{|\mathcal{V}|}{|\mathcal{L}|} = 0.9$. The lower success rate in this case implies more fallbacks, which reduces the net speed gains.
>
> ---
> ### (3) Benefit in real-world applications
> > the scope of this work might be a bit narrow.
>
> > the experiments are for fairly small number of variables and degree.
>
> > It remains unclear which real-world applications benefit from these runtime improvements on such small systems.
>
> We currently focus on relatively small systems, and the immediate real-world impact may be limited. However, we kindly ask the reviewer to evaluate our contributions within the context of prior studies on learning Gröbner and border basis computation - our study handles the largest and the most general class of polynomial systems.
>
> - [Peifer+'20] handles binomial systems (both input F and output G) with n <= 5 (mainly n = 3).
> - [Kera+'24] handles up to n = 5, the mixing matrix A in F = AG is sparsified for the manageable input size (below 5000 tokens).
>
> The table below compares the average size of input systems F for n = 5 (numbers are rounded):
>
> |                | Input system size \|F\| | Avg # of terms in F | class of ideals and systems | Note                                                  |
> |----------------|-------------------------|----------------------|----------------------| -------------------------------------------------------|
> | [Peifer+’20]   | 10                      | 20                   | binomial |  Most experiments were done with n=3.
> | [Kera+'24]     | 5–7                     | 42                   | shape position (\|F\| >= \|G\| = n) | From "Max terms in F" in Table 6 in [Kera+'24].       |
> | Ours           | 6–10                    | 130                  | vanishing ideal (\|G\| >= n, OK with \|F\| < \|G\|) | Computed from our datasets, will be added in Appendix.|
>
> Solving large polynomial systems is computationally challenging, and deep learning–based approaches have recently emerged as a promising alternative. Unlike [Peifer+'20], we target general polynomial systems (not just binomials), and unlike [Kera+'24], our method guarantees correctness.
>
> Moreover, our **monomial embedding** enables efficient encoding of large polynomial systems for Transformer models, reducing input size by a factor of $n + 2$, and memory usage by $O((n + 2)^2)$. and our **ideal-invariant transform** (Thm. 4.5) provides a principled method for generating random non-trivial systems.
>
> Our work makes substantial contributions across methodology, deep learning techniques, and dataset construction - all of which we believe advance deep learning–based approaches toward solving larger and more general polynomial systems.
>
> ---
>
> ### Others
> > how you chose the Transformer architecture [...] any ablation about that?
>
> We adopted a standard architecture from [Vaswani+'17], which aligns with [Kera+'24]. In our preliminary experiments, the number of layers and embedding dimensions had little impact on learning, so we decided to focus on this standard model (with monomial embedding). The ablation of monomial embedding is infeasible, as the input size is too large without it. Instead, Tab. 3 provides ablation with another task, demonstrating that monomial embedding improves training cost and final success rate significantly.
>
> > The algorithm proposed is limited to finite-sized polynomial systems.
>
> The reviewer may intend "finite-field." Note that this is not a critical limitation. From an algebraic perspective, it is common to solve infinite-field systems through finite-field ones using *modular technique* [Arnold'03]. From a learning perspective, learning in an infinite field is far easier, as many studies have reported. We kindly ask the reviewer to consult our response to Reviewer C42S ("Finite field only").
>
> ---
> - [Peifer+'20] Peifer et al., "Learning Selection Strategies in Buchberger's Algorithm," ICML'20
> - [Kera+'24] Kera et al., "Learning to Compute Gröbner bases," NeurIPS'24
> - [Vaswani+'17] Vaswani et al., "Attention is All You Need," NeurIPS'17
> - [Arnold'03]  Arnold, "Modular algorithms for computing Gröbner bases," JSC, 2003

---

> ### Author Response · Authors · 2025-08-06
> **Discussion Closing**
>
> Dear Reviewer,
>
> We kindly remind you that the discussion period is coming to a close. We believe that our response addresses your concerns, particularly by clarifying several potential misconceptions and providing clearer contrasts with related work.
>
> We understand that our rebuttal is quite detailed, so we would sincerely appreciate it if you could take a moment to review at least the **Key Responses** section.
>
> Given that you are currently the *only* reviewer recommending rejection, we would greatly appreciate the opportunity to continue the discussion with you before the post-rebuttal reviewer phase begins. We believe that clarifying any remaining concerns now would help ensure a more efficient and focused discussion among reviewers later.
>
> Thank you very much for your time and consideration.

---

> ### Author Response · Authors · 2025-08-08
> **Official Comment by Authors**
>
> We are pleased to hear that you recognize the technical novelty and experimental results of our work and will be revisiting the evaluation. Once again, we appreciate your time, feedback, and discussion.

---

### Note · Authors · 2025-08-12

We would like to express our sincere gratitude to the reviewers and the AC for their dedication in evaluating our work. We are pleased that the careful reviews and fruitful discussions have led all reviewers to a *unanimous* recommendation for acceptance.

Below, we summarize the key points from the rebuttal and discussion process to assist the AC in making the final decision.

---

**Summary of This Work**

This work proposes the Oracle Border Basis Algorithm for solving polynomial systems. Compared to prior studies:

- It addresses the largest and most general polynomial systems to date (cf. any table in the rebuttal).
- Unlike recent end-to-end Transformer-based approaches, ours integrates the Transformer into an algebraic algorithm, thereby guaranteeing the correctness of the output bases.

Several technical contributions—both algebraic and machine learning—are crucial to achieving this. In particular, the following two extend beyond border basis computation:

- **Monomial Embedding.** Reduces the number of input tokens by a factor of *n* and memory cost by *n²*. We also observe improved success rates in cumulative product tasks.
- **Ideal-Invariant Generator Transform.** Provides theoretical support for simple sampling of a “mixing matrix” *A* that preserves the ideal before and after transforming the generating set. At a high level, this yields a generic way to sample generators of zero-dimensional systems.

---

**Resolution Through Rebuttal and Discussion**

Reviewers appreciated the technical contributions. Concerns focused on (1) the problem setup (zero-dimensional, finite-field systems), (2) the experimental setup and results, and (3) the real-world impact. The first two largely stemmed from understatements or misconceptions. In brief:

1. Positive-dimensional, infinite-field systems can be reduced to our case (zero-dimensional, finite-field).
2. The experiments comprise 27 datasets for training/evaluation and **27 additional sets for out-of-distribution evaluation** (the latter was overlooked). Across these diverse case studies, we observed empirical gains of up to **30 seconds** (misunderstood as ~0.1s) with strong generalization.
3. While the immediate real-world impact may be limited, our technical advances and empirical results represent a significant step toward broader applicability.

We will revise the paper to make these points clearer, and we hope these final remarks assist the AC in making the final decision.

---

### Decision · Program_Chairs · 2025-09-17

**Decision:**

Accept (poster)

**Comment:**

The paper is a borderline accept. While reviewers initially had concerns about scalability, originality, and practical applicability, the authors' detailed rebuttal successfully clarified these points. They demonstrated technical novelty, significant real-world speedups (30+ seconds), and strong generalization to out-of-distribution instances, convincing all reviewers to recommend acceptance.